# BA-LoRA: Bias-Alleviating Low-Rank Adaptation to Mitigate Catastrophic Inheritance in Large Language Models

## Abstract

Large language models (LLMs) have demonstrated remarkable proficiency across various natural language processing (NLP) tasks. However, adapting LLMs to downstream applications requires computationally intensive and memory-demanding fine-tuning procedures. To alleviate these burdens, parameter-efficient fine-tuning (PEFT) techniques have emerged as a promising approach to tailor LLMs with minimal computational overhead. While PEFT methods offer substantial advantages, they do not fully address the pervasive issue of bias propagation from pre-training data. This work introduces Bias-Alleviating Low-Rank Adaptation (BA-LoRA), a novel PEFT method designed to counteract bias inheritance. BA-LoRA incorporates three distinct regularization terms: (1) a consistency regularizer, (2) a diversity regularizer, and (3) a singular value decomposition regularizer. These regularizers aim to enhance the models' consistency, diversity, and generalization capabilities during fine-tuning. We conduct extensive experiments on natural language understanding (NLU) and natural language generation (NLG) tasks using prominent LLMs such as LLaMA, Mistral, and Gemma. The results demonstrate that BA-LoRA outperforms LoRA and its state-of-the-art variants. Moreover, our method effectively mitigates the adverse effects of pre-training bias, leading to more reliable and robust model outputs.

## 1 Introduction

The emergence of large language models (LLMs) has marked a new era in natural language processing (NLP). Models such as GPT-4 (OpenAI, 2023), Llama (Touvron et al., 2023), Mistral (Jiang et al., 2023), and Gemma (Team et al., 2024) have demonstrated exceptional performance across a wide array of NLP tasks, including language comprehension, generation, and reasoning (Zhao et al., 2023; Chang et al., 2024). The remarkable advancements of LLMs can be largely attributed to their training on vast datasets (Zhao et al., 2023). As LLMs continue to evolve rapidly, training on extensively scaled web-derived corpora has become standard practice to improve model generalization, thus bypassing the labor-intensive processes of data curation and annotation (Gao et al., 2020; Penedo et al., 2023). However, the corresponding increase in data volume has introduced several challenges, such as the presence of imbalanced, duplicated, and corrupted information (Parashar et al., 2024; Liu & He, 2024; Chen et al., 2024b; Yang et al., 2023).

Recent research has shown that various forms of bias in training data can negatively affect LLM behavior (Dong et al., 2023; Dodge et al., 2021; Longpre et al., 2023; Chen et al., 2024a). For example, noise within the training data can degrade model generalization (Chen et al., 2024a), while the long-tailed distribution of concepts in web-scale data can cause LLMs to overemphasize overrepresented topics (Zhu et al., 2024). Furthermore, biases introduced during pre-training can persist even after fine-tuning, potentially compromising model performance and safety in real-world applications (Qi et al., 2023; Bommasani et al., 2021; Mallen et al., 2022; Carlini et al., 2023).

This phenomenon, termed "Catastrophic Inheritance" by (Chen et al., 2024a), has spurred investigations into mitigation strategies. While constructing less biased datasets and developing more robust model architectures are prominent approaches (Liu & He, 2024), this study explores an alternative: innovations in fine-tuning. Fine-tuning LLMs is a powerful method for enhancing task-specific

performance (Han et al., 2024), aligning models with user intent (Ouyang et al., 2022; Xu et al., 2024), and eliciting desired behaviors (Bai et al., 2022; Rafailov et al., 2024). However, fine-tuning large-scale models' computational and memory demands are substantial (Hu et al., 2021). For instance, 16-bit fine-tuning of a Llama-65B model requires over 780 GB of GPU memory (Dettmers et al., 2024). To address these limitations, parameter-efficient fine-tuning (PEFT) techniques, such as Low-Rank Adaptation (LoRA) (Hu et al., 2021), have gained prominence.

LoRA posits that parameter updates during fine-tuning can be efficiently represented by low-rank matrices. Therefore, given a pre-trained weight matrix $W \in \mathbb{R}^{m \times n}$, instead of updating all parameters of $W$ directly, LoRA introduces an auxiliary low-rank adapter $\Delta W = AB$, where $A \in \mathbb{R}^{m \times r}$ and $B \in \mathbb{R}^{r \times n}$ with rank $r \ll \min(m, n)$. Here, $A$ and $B$ are learnable matrices initialized as:

- $A$ is initialized with a scaled normal distribution: $a_{ij} \sim \mathcal{N}(0, \sigma^2)$.
- $B$ is initialized to zero: $b_{ij} = 0$.

For a given input $X$, the output $Y$ is computed as:

$$Y = X(W + \Delta W) = X(W + AB). \tag{1}$$

Only $A$ and $B$ are updated during fine-tuning while $W$ remains frozen. This initialization ensures that $AB = 0$ at the start of training, thus preserving the model's original output. Since the rank $r$ is significantly smaller than the dimensions of $W$, LoRA substantially reduces training overhead compared to full fine-tuning (Hu et al., 2021).

To mitigate the detrimental effects of Catastrophic Inheritance, particularly noise and imbalance, we propose Bias-Alleviating Low-Rank Adaptation (BA-LoRA). Building upon Principal Singular Values and Singular Vectors Adaptation (PiSSA) (Meng et al., 2024), which addresses convergence issues in standard LoRA, our approach incorporates three distinct regularization terms: a consistency regularizer, a diversity regularizer, and a singular value decomposition (SVD) regularizer. The consistency regularizer preserves valuable pre-trained knowledge during fine-tuning, while the diversity regularizer encourages varied model outputs. The SVD regularizer enhances the generalization capabilities of generative models. Recognizing the fundamental differences between Natural Language Understanding (NLU) and Natural Language Generation (NLG), such as determinism in NLU versus diversity in NLG, we tailor our regularization strategies accordingly.

To evaluate the efficacy of BA-LoRA, we conduct comprehensive experiments across diverse benchmarks, including mathematical reasoning (GSM8K (Cobbe et al., 2021) and MATH (Yu et al., 2023)), coding (HumanEval (Chen et al., 2021) and MBPP (Austin et al., 2021)), natural language understanding (GLUE (Wang et al., 2018)), and general language evaluation (MT-Bench (Zheng et al., 2024)). Our experiments utilize prominent LLMs such as LLaMA 2-7B (Touvron et al., 2023), Mistral-7B (Jiang et al., 2023), and Gemma-7B (Team et al., 2024), as well as encoder-only architectures like RoBERTa-large (Liu et al., 2019) and DeBERTa-v3-base (He et al., 2021b). The results unequivocally demonstrate BA-LoRA's superiority over LoRA and PiSSA. Moreover, our method effectively attenuates noise inherited from pre-training, leading to more robust and generalizable models.

## 2 RELATED WORKS

Parameter-efficient fine-tuning (PEFT) techniques (Xu et al., 2023b; Han et al., 2024) have garnered significant attention as an approach to adapting LLMs for specific tasks under limited hardware resources. Three main categories of PEFT techniques are commonly used. The first category includes adapter-based methods (Houlsby et al., 2019b; Lin et al., 2020; Lei et al., 2023; He et al., 2021a), which introduce additional layers into the model and fine-tune these layers (typically with far fewer parameters) to reduce computational costs. The second category comprises soft prompt tuning methods (Hambardzumyan et al., 2021; Lester et al., 2021; Li & Liang, 2021a; Liu et al., 2023), which prepend learnable soft prompts to the model's input to tailor it to specific tasks. These methods leverage the inherent capabilities of pre-trained models, requiring only the appropriate prompts to adapt to downstream tasks. The third category encompasses low-rank adaptation (LoRA) and its

variants (Hu et al., 2021; Zhang et al., 2022; Dettmers et al., 2024). LoRA introduces the product of low-rank matrices within existing layers to approximate weight updates during fine-tuning (Hu et al., 2021).

Variants of LoRA enhance its efficiency and performance in different ways. AdaLoRA adaptively distributes the parameter budget among weight matrices based on their importance, improving efficiency and performance by pruning unimportant updates and minimizing computational overhead (Zhang et al., 2022). DoRA increases LoRA's learning capacity and stability by decomposing pre-trained weights into magnitude and direction components for fine-tuning (Liu et al., 2024). LoHA enhances LoRA by employing Hamiltonian products (Hyeon-Woo et al., 2021). DyLoRA addresses the fixed size and rank optimization limitations of LoRA by dynamically training LoRA blocks across varying ranks (Valipour et al., 2022). DeltaLoRA improves the representational capacity of LoRA by updating the model's original weights using parameters from adapter layers (Zi et al., 2023). PiSSA initializes adapter matrices $A$ and $B$ to approximate the original matrix $W$ through singular value decomposition, leading to faster convergence and improved performance (Meng et al., 2024). While many LoRA variants focus on accelerating convergence or reducing memory consumption, our BA-LoRA method uniquely addresses the core challenge of Catastrophic Inheritance in LLM fine-tuning.

## 3 METHOD

### 3.1 PRINCIPAL SINGULAR VALUES AND SINGULAR VECTORS ADAPTATION (PiSSA)

As a variant of LoRA, PiSSA addresses the convergence speed challenge by retaining the core LoRA architecture while innovating in initialization. Specifically, PiSSA leverages the principal components of the original weight matrix, $W$, to initialize the adapter matrices, $A$ and $B$. The remaining components are encapsulated within a residual matrix, $W^{res} \in \mathbb{R}^{m \times n}$. The SVD of $W \in \mathbb{R}^{m \times n}$ is expressed as $W = USV^T$, where $U \in \mathbb{R}^{m \times \min(m,n)}$ and $V \in \mathbb{R}^{n \times \min(m,n)}$ are orthogonal singular vectors, and $S = diag(s) \in \mathbb{R}^{\min(m,n) \times \min(m,n)}$ is a diagonal matrix, where the operation $diag(s)$ transforms $s$ to $S$ and $s \in \mathbb{R}^{\min(m,n)}_{\leq 0}$ represents the singular values arranged in descending order. PiSSA partitions the singular values and vectors into principal and residual components, denoted as $\{U_{[:,:r]}, S_{[:r,:r]}, V_{[:,:r]}\}$ and $\{U_{[:,r:]}, S_{[r:,r:]}, V_{[:,r:]}\}$, respectively, where the matrix slicing notations are the same as those in PyTorch, $[:r]$ denotes the first $r$ dimensions, and $r$ signifies the intrinsic rank of $W$. The principal components are then employed to initialize the low-rank adapter with $A \in \mathbb{R}^{m \times r}$ and $B \in \mathbb{R}^{r \times n}$:

$$A = U_{[:,:r]} S_{[:r,:r]}^{1/2} \in \mathbb{R}^{m \times r}, \tag{2}$$

$$B = S_{[:r,:r]}^{1/2} V_{[:,:r]}^T \in \mathbb{R}^{r \times n}. \tag{3}$$

The residual matrix $W^{res}$ remains frozen during fine-tuning:

$$W^{res} = U_{[:,r:]} S_{[r:,r:]} V_{[:,r:]}^T \in \mathbb{R}^{m \times n}. \tag{4}$$

PiSSA preserves the pre-trained model's full capacity at the start of fine-tuning by using $W = W^{res} + AB$. This approach prioritizes training the most influential parameters, thereby accelerating convergence. Inheriting LoRA's benefits of reduced parameter count and deployment simplicity, PiSSA further leverages efficient SVD computations to expedite the training process.

### 3.2 BIAS-ALLEVIATING LOW-RANK ADAPTATION (BA-LoRA)

Catastrophic Inheritance encapsulates the challenges posed by biased large-scale training data, which can manifest in LLMs as vulnerabilities and limitations arising from duplicated, noisy, imbalanced, or unethical samples. These inherited flaws can adversely impact downstream tasks, leading to diminished generalization, degraded performance, security breaches, and biased outputs. To address the specific issues caused by noisy and imbalanced data, we introduce BA-LoRA, a method incorporating three distinct regularization terms: (1) consistency regularizer, (2) diversity regularizer, and (3) SVD

regularizer. Recognizing the nuanced differences between NLU and NLG, we have tailored specific variants of each regularizer to optimize performance for respective task domains.

### 3.2.1 REGULARIZATIONS FOR NLU TASKS

**Consistency Regularization.** To safeguard valuable pre-trained knowledge during the fine-tuning process, we introduce a regularization term based on the mean squared error (MSE) loss between normalized output logits produced by the pre-trained model, $\mathbf{F}_P$, and those generated by the fine-tuned model, $\mathbf{F}_F$. This loss function incentivizes the fine-tuned model to retain essential pre-trained information while adapting to downstream task requirements.

$$\mathcal{L}_{\text{CR\_NLU}} = \left\| \frac{\mathbf{F}_p}{\|\mathbf{F}_p\|_2} - \frac{\mathbf{F}_f}{\|\mathbf{F}_f\|_2} \right\|_2^2 \tag{5}$$

This objective facilitates the inheritance of critical pre-trained knowledge in $\mathbf{F}_f$ after fine-tuning.

**Diversity Regularization.** To address the detrimental effects of imbalanced data, we introduce a diversity regularizer aimed at eliciting more diverse representational structures within LLMs and preventing the encoding of semantically similar samples during fine-tuning. Inspired by (Bardes et al., 2021), we employ a covariance loss to minimize the off-diagonal elements of the covariance matrix of the fine-tuned outputs $\mathbf{F}_f$:

$$\mathcal{L}_{\text{DR\_NLU}} = \frac{1}{D} \sum_{i \neq j} [C(\mathbf{F}_f)]_{i,j}^2 \tag{6}$$

where $D$ represents the dimensionality of $\mathbf{F}_f$ and $C(\mathbf{F}_f)$ is the covariance matrix of $\mathbf{F}_f$, which is defined as:

$$C(\mathbf{F}_f) = \frac{1}{M-1} \sum_{i=1}^{M} \left( f_i - \bar{f} \right) \left( f_i - \bar{f} \right)^T \tag{7}$$

where $M$ denotes the number of elements involved in $\mathbf{F}_f$, $f_i$ is the $i$-th element in $\mathbf{F}_f$, and $\bar{f}$ is the mean value of $\mathbf{F}_f$.

**Singular Value Decomposition Regularization.** The SVD regularizer is designed to enhance model generalizability to mitigate the adverse effects of noisy data. Building upon the insight from (Chen et al., 2019) that eigenvectors corresponding to the largest singular values significantly contribute to model generalizability, we propose an SVD regularizer that maximizes the sum of the top $k$ singular values of a batched fine-tuned output matrix:

$$\mathcal{L}_{\text{SVDR\_NLU}} = -\frac{\sum_{i=1}^{k} \sigma_i}{\sum_{j=1}^{D} \sigma_j} \tag{8}$$

where $k$ is a hyperparameter, $\sigma_i$ denotes the $i$-th singular value of the top $k$ singular values of the output matrix, and $\sum_{j=1}^{D} \sigma_j$ is the sum of all singular values obtained from the SVD of the output matrix. This decomposition represents the matrix as $\mathbf{U}\mathbf{\Sigma}\mathbf{V}^{\top}$, where $\mathbf{\Sigma}$ is a diagonal matrix containing singular values $\{\sigma_1, \ldots, \sigma_D\}$. This regularization term emphasizes significant components of the logit matrix, enhancing the model's generalizability across various downstream tasks.

### 3.2.2 OVERALL OBJECTIVE FUNCTION FOR NLU

The overall objective function for NLU tasks is formulated as follows:

$$\mathcal{L}_{\text{NLU}} = \mathcal{L}_{\text{task\_NLU}} + \lambda_1 \mathcal{L}_{\text{CR\_NLU}} + \lambda_2 \mathcal{L}_{\text{DR\_NLU}} + \lambda_3 \mathcal{L}_{\text{SVDR\_NLU}} \tag{9}$$

where $\mathcal{L}_{\text{task\_NLU}}$ represents the standard cross-entropy loss function for the downstream task, and $\lambda_1$, $\lambda_2$, and $\lambda_3$ are weighting parameters to balance each regularization term.

### 3.2.3 REGULARIZATIONS FOR NLG TASKS

**Consistency Regularization.** To ensure that the fine-tuned model retains knowledge from pre-training, we utilize the Kullback-Leibler Divergence (KLD) to measure the divergence between the output distributions of the fine-tuned and pre-trained models (Dong et al., 2021). Specifically, we define the consistency regularization loss as:

$$\mathcal{L}_{\text{CR\_NLG}} = \frac{1}{T} \sum_{t=1}^{T} \text{KL} \left( \mathcal{P}_{\text{pt}}(y_t \mid y_{<t}, x) \parallel \mathcal{P}_{\text{ft}}(y_t \mid y_{<t}, x) \right), \tag{10}$$

where $\mathcal{P}_{\text{pt}}(y_t \mid y_{<t}, x)$ and $\mathcal{P}_{\text{ft}}(y_t \mid y_{<t}, x)$ represent the conditional probability distributions of the pre-trained model and the fine-tuned model, respectively. For the current token $y_t$, given the input $x$ and the preceding token sequence $y_{<t}$. KLD encourages the model to continuously retain useful pre-training information during the fine-tuning process, which is crucial for maintaining the style and coherence of the generation task.

**Diversity Regularization.** To enhance the diversity of the generated text, we introduce an entropy-based regularization term, inspired by previous work (Gat et al., 2020). This regularization term aims to increase the entropy of the predicted token distributions during fine-tuning, thus encouraging more varied and diverse outputs.

$$\mathcal{L}_{\text{DR\_NLG}} = -\frac{1}{T} \sum_{t=1}^{T} \sum_{i=1}^{N} P_{\text{ft}}(x_i|h_t) \log P_{\text{ft}}(x_i|h_t) \tag{11}$$

where, $P_{\text{ft}}(x_i|h_t)$ represents the probability assigned to token $x_i$ at time step $t$, given the model's hidden state $h_t$. Maximizing entropy at each time step minimizes repetitive outputs and encourages more diverse and enriched text generation.

**Singular Value Decomposition Regularization** To enhance the generalization capability of generative models, we introduce a regularization technique that accentuates the most significant singular values, enabling the model to capture the principal components of the data and prioritize the most informative patterns.

$$\mathcal{L}_{\text{SVDR\_NLG}} = -\frac{\sum_{i=1}^{k} \sigma_i}{\sum_{j=1}^{D} \sigma_j} \tag{12}$$

where $\sigma_i$ denotes the $i$-th largest singular value, $\sigma_j$ represents the $j$-th singular value, and $D$ is the total number of singular values. This regularization term aims to maximize the relative contribution of the top $k$ singular values, encouraging the model to focus on the most critical aspects of the data. By integrating this regularization, the model is steered towards generating higher-quality and more diverse outputs, ultimately improving its performance and robustness.

### 3.2.4 OVERALL OBJECTIVE FUNCTION FOR NLG

The objective function for downstream NLG tasks is formulated as follows:

$$\mathcal{L}_{\text{NLG}} = \mathcal{L}_{\text{task\_NLG}} + \lambda_1 \mathcal{L}_{\text{CR\_NLG}} + \lambda_2 \mathcal{L}_{\text{DR\_NLG}} + \lambda_3 \mathcal{L}_{\text{SVDR\_NLG}} \tag{13}$$

where $\mathcal{L}_{\text{task\_NLG}}$ denotes the standard loss for the downstream generative task, and $\lambda_1$, $\lambda_2$, and $\lambda_3$ are weighting parameters to balance each regularization term.

## 4 EXPERIMENTS

This section presents a comprehensive evaluation of our proposed BA-LoRA method across a diverse range of natural language generation (NLG) and natural language understanding (NLU) benchmarks. Our results unequivocally demonstrate the superiority of BA-LoRA over existing LoRA variants.

Furthermore, through rigorous experimentation, we elucidate BA-LoRA's efficacy in mitigating the adverse impacts of noisy data, thereby enhancing model robustness and generalizability.

## 4.1 MODELS AND DATASETS

To evaluate the effectiveness of our approach, we conduct experiments using several prominent language models and assess their performance on a diverse array of datasets, covering both **Natural Language Generation** and **Natural Language Understanding** tasks. Specifically, for language generation models, we include LLaMA 2-7B (Touvron et al., 2023), LLaMA 3-8B (AI@Meta, 2024), Mistral-7B (Jiang et al., 2023), Gemma-7B (Team et al., 2024), and GPT-2-XL (Radford et al., 2019). For language understanding models, we use BERT-Large (BERT-L) (Devlin et al., 2018), and DeBERTa-v3-base (He et al., 2021b). This selection ensures coverage across various architectures and parameter scales, facilitating a comprehensive evaluation.

For the datasets, we employ a wide range of tasks in both **Natural Language Generation** (GSM8K (Cobbe et al., 2021), MATH (Yu et al., 2023), HumanEval (Chen et al., 2021), MBPP (Austin et al., 2021), MT-Bench (Zheng et al., 2024)) and **Natural Language Understanding**. In the latter, we assess in-domain (ID) performance using the GLUE benchmark (Wang et al., 2018) and out-of-domain (OOD) generalization using the GLUE-X benchmark (Yang et al., 2022). These datasets span a broad range of challenges, allowing for a thorough examination of our method's generalization capabilities.

## 4.2 IMPLEMENTATION DETAILS

In our experiments, we adopt the PiSSA (Meng et al., 2024) implementation strategy. We compute the loss using only the responses from the instruction-following dataset, ensuring lora_dropout to 0. We utilize the Float32 computation type for both the base model and the adapter in BA-LoRA. For the NLU tasks, we set the hyperparameters as: $\lambda_1 = 1e-4$, $\lambda_2 = 4e-4$, and $\lambda_3 = 1e-4$. We set lora_r = lora_alpha = 128 and use AdamW (Loshchilov & Hutter, 2017) optimizer with a batch size of 128, a learning rate of $2e-5$, cosine annealing schedules, and a warmup ratio of 0.03, without any weight decay. For the NLG tasks, the hyperparameters are set as: $\lambda_1 = 1e-4$, $\lambda_2 = 3e-4$, and $\lambda_3 = 1e-4$. We set lora_r as 8 and select lora_alpha in 8, 16. We utilize AdamW with a linear learning rate schedule to optimize and tune the learning rate (LR) from $1e-4$, $2e-4$, $3e-4$, $4e-4$, $5e-4$, $6e-4$, $5e-5$, $3e-5$. Batch sizes (BS) are selected from 6, 8, 16, 32. Appendix Section B presents the detailed hyperparameters we utilized on the GLUE benchmark. comparison. All experiments were conducted using NVIDIA A40 (48G) GPUs. All presented results are derived from three independent experiments, ensuring the consistency and robustness of our findings. Each experiment was conducted under identical conditions, and the results were averaged to mitigate any variability. We enhanced $\mathcal{L}_{\text{SVDR\_NLG}}$'s computational efficiency by using partial SVD to compute only the top $k$ singular values, reducing overhead.

## 4.3 RESULTS AND ANALYSIS

Table 1: Performance Comparison of Various Models and Methods on NLG Tasks. The best and second-best results are highlighted in **bold** and underline.

| Models | Methods | Parameters | GSM8K | MATH | HumanEval | MBPP | MT-Bench | Avg |
|--------|---------|------------|-------|------|-----------|------|----------|-----|
| LLaMA-2-7B | Full FT | 6738M | 49.05 | 7.22 | 21.34 | 35.59 | 4.91 | 23.62 |
| | LoRA | 320M | 42.47 | 5.60 | 17.03 | 31.48 | 4.62 | 20.24 |
| | PiSSA | 320M | 52.01 | 7.76 | 21.55 | 33.09 | 4.87 | 23.86 |
| | BA-LoRA | 320M | **53.83** | **9.13** | **23.58** | **36.86** | **5.11** | **25.70** |
| Mistral-7B | Full FT | 6738M | 67.02 | 18.60 | 45.12 | 51.38 | 4.95 | 37.41 |
| | LoRA | 168M | 67.68 | 19.90 | 42.54 | 55.74 | 4.92 | 38.16 |
| | PiSSA | 168M | 71.90 | 21.72 | 45.20 | 60.83 | 5.23 | 40.98 |
| | BA-LoRA | 168M | **73.04** | **22.11** | **46.31** | **62.29** | **5.41** | **41.83** |
| Gemma-7B | Full FT | 6738M | 71.34 | 22.74 | 46.95 | 55.64 | 5.40 | 40.41 |
| | LoRA | 200M | 74.64 | 31.16 | 51.64 | 62.84 | 5.01 | 45.06 |
| | PiSSA | 200M | 77.58 | 31.47 | 53.15 | 65.49 | 5.66 | 46.67 |
| | BA-LoRA | 200M | **78.13** | **32.25** | **54.41** | **66.12** | **5.73** | **47.33** |

Table 2: Performance comparison of different baseline methods on NLU tasks. The best and second-best results are highlighted in **bold** and underline.

| Methods | MNLI | SST-2 | MRPC | CoLA | QNLI | QQP | RTE | SST-B | Avg |
|---------|------|-------|------|------|------|-----|-----|-------|-----|
| Full FT | 89.90 | 95.61 | 89.50 | 69.23 | 94.09 | 92.44 | 83.85 | 91.71 | 88.29 |
| BitFit | 89.37 | 94.84 | 87.75 | 66.96 | 92.24 | 88.41 | 78.70 | 91.35 | 86.20 |
| HAdapter | 90.13 | 95.53 | 89.95 | 68.64 | 94.11 | 91.91 | 84.48 | 91.48 | 88.28 |
| PAdapter | 90.33 | 95.61 | 89.46 | 68.77 | 94.29 | 92.04 | 85.20 | 91.60 | 88.41 |
| LoRA | 90.71 | 94.79 | 89.85 | 70.05 | 93.94 | 92.07 | 85.43 | 91.67 | 88.56 |
| LoHA | 90.74 | 94.92 | 90.43 | 70.63 | 93.95 | 92.05 | 86.41 | 91.72 | 88.86 |
| DoRA | 90.29 | 95.79 | 90.93 | 70.85 | 94.10 | 92.17 | 86.04 | 91.79 | 89.00 |
| DyLoRA | 90.97 | 95.21 | 91.45 | 70.79 | 94.08 | 92.29 | 86.57 | 91.86 | 89.15 |
| AdaLoRA | 90.76 | 96.10 | 90.69 | 71.45 | 94.55 | 92.23 | 87.59 | 91.84 | 89.40 |
| PiSSA | 90.47 | 95.81 | 91.48 | 72.27 | 94.41 | 92.21 | 87.14 | 91.93 | 89.47 |
| **BA-LoRA** | **90.92** | **96.25** | **91.83** | **72.79** | **94.84** | **92.59** | **87.87** | **92.15** | **89.91** |

### 4.3.1 ANALYSIS OF THE NLG AND NLU PERFORMANCE OF BA-LORA

To evaluate BA-LoRA's effectiveness on NLG tasks, we fine-tuned LLaMA-2-7B, Mistral-7B, and Gemma-7B on the MetaMathQA dataset (Yu et al., 2023) and assessed their mathematical problem-solving capabilities using the GSM8K (Cobbe et al., 2021) and MATH (Yu et al., 2023) validation sets, reporting Accuracy. Similarly, models were fine-tuned on the CodeFeedback dataset (Zheng et al., 2024) and evaluated for coding proficiency via HumanEval (Chen et al., 2021) and MBPP (Austin et al., 2021), with PASS@1 metrics reported. To assess conversational abilities, models were trained on the WizardLM-Evol-Instruct dataset (Xu et al., 2024) and evaluated on MT-Bench (Zheng et al., 2024), with response quality judged by GPT-4 and first turn scores reported. All experiments utilized 100K data points and a single training epoch for efficiency.

Table 1 presents the experimental outcomes, clearly demonstrating BA-LoRA's superior performance compared to baseline methods. For instance, BA-LoRA enhanced LLaMA 2-7B, Mistral-7B, and Gemma-7B performance on GSM8K by 1.82%, 1.14%, and 0.55%, respectively, compared to PiSSA. HumanEval improvements were 2.03%, 1.11%, and 1.26%, while MT-Bench enhancements reached 0.24%, 0.18%, and 0.07%. Notably, BA-LoRA achieved a remarkable 6.92% performance uplift over full parameter fine-tuning on Gemma, utilizing only 2.3% of trainable parameters across five tasks.

To assess the effectiveness of BA-LoRA on natural language understanding (NLU) tasks, we conducted experiments on the GLUE benchmark (Wang et al., 2018), which includes two single-sentence classification tasks (CoLA, SST), five paired-text classification tasks (MNLI, RTE, QQP, MRPC, QNLI), and one text similarity prediction task (STS-B). The evaluation metrics comprise the overall matched and mismatched accuracy for MNLI, the Matthews correlation coefficient for CoLA, the Pearson correlation coefficient for STS-B, and accuracy for the remaining tasks. We used the DeBERTa-v3-base model (He et al., 2021b) and compared BA-LoRA against ten baseline methods, including Full Fine-Tuning (Full FT), BitFit (Zaken et al., 2021), HAdapter (Houlsby et al., 2019a), PAdapter (Pfeiffer et al., 2020), LoRA (Hu et al., 2021), LoHA (Hyeon-Woo et al., 2021), DoRA (Liu et al., 2024), DyLoRA (Valipour et al., 2022), AdaLoRA (Zhang et al., 2022), and PiSSA (Meng et al., 2024). Table 2 presents the results of DeBERTa-v3-base across eight tasks, demonstrating the consistent superiority of BA-LoRA over all baselines. On average, BA-LoRA outperforms PiSSA and LoRA by 0.44% and 1.35%, respectively. These results underscore the effectiveness of BA-LoRA in enhancing the performance of NLU models.

A comparative analysis of Tables 1 and 2 reveals BA-LoRA's consistent performance advantages across both NLG and NLU tasks. This indicates BA-LoRA's proficiency in augmenting both generative and comprehension capabilities for language models. By incorporating consistency, diversity, and SVD regularization, BA-LoRA effectively mitigates the adverse effects of Catastrophic Inheritance, fostering consistent, diverse, and generalized model outputs. Furthermore, BA-LoRA's modest computational requirements render it suitable for efficient fine-tuning of LLMs with limited resources.

### 4.3.2 ANALYSIS ON MITIGATE NOISY DATA

This study aims to evaluate BA-LoRA's efficacy in mitigating the detrimental effects of noise inherent in large-scale pre-training data on downstream tasks. Given the ubiquitous presence of noise in

Table 3: ID Performance Comparison of BERT-L and GPT-2-XL Using LoRA and BA-LoRA Methods on GLUE Benchmark. The best outcome is highlighted in **bold**.

| Model | Methods | MNLI | SST-2 | MRPC | CoLA | QNLI | QQP | RTE | STS-B | Avg |
|-------|---------|------|-------|------|------|------|-----|-----|-------|-----|
| BERT-L | LoRA | 87.24 | 93.19 | 90.10 | 64.73 | 93.13 | 90.94 | 73.14 | 90.63 | 85.39 |
| | BA-LoRA | **89.72** | **94.85** | **92.23** | **65.49** | **95.48** | **91.72** | **75.77** | **91.71** | **87.12** |
| GPT-2-XL | LoRA | 85.28 | 95.38 | 86.17 | 50.63 | 89.42 | 88.56 | 72.29 | 89.27 | 82.13 |
| | BA-LoRA | **88.14** | **96.52** | **89.23** | **52.76** | **91.26** | **89.95** | **74.57** | **90.83** | **84.16** |

Table 4: OOD Performance Comparison of BERT-L and GPT-2-XL Using LoRA and BA-LoRA Methods on GLUE-x Benchmark. The best outcome is highlighted in **bold**.

| Model | Methods | MNLI | SST-2 | MRPC | CoLA | QNLI | QQP | RTE | STS-B | Avg |
|-------|---------|------|-------|------|------|------|-----|-----|-------|-----|
| BERT-L | LoRA | 85.19 | 93.49 | 89.93 | 63.49 | 92.32 | 87.73 | 73.65 | 90.57 | 84.55 |
| | BA-LoRA | **87.91** | **94.18** | **90.62** | **65.81** | **93.04** | **89.06** | **75.41** | **91.21** | **85.91** |
| GPT-2-XL | LoRA | 87.02 | 95.11 | 86.81 | 60.95 | 91.77 | 87.59 | 78.76 | 89.25 | 84.66 |
| | BA-LoRA | **89.58** | **96.40** | **88.18** | **63.11** | **92.68** | **88.62** | **81.21** | **90.37** | **86.27** |

human-annotated datasets, its influence on pre-training is unavoidable. To comprehensively assess the impact of noisy pre-training data, we employ both ID and OOD evaluation using the GLUE and GLUE-x benchmarks, respectively. BERT-L (Devlin et al., 2018), pre-trained on BooksCorpus (Zhu et al., 2015) and English Wikipedia, and GPT-2-XL (Radford et al., 2019), pre-trained on the noisy WebText dataset derived from Common Crawl, serve as our models.

As detailed in Tables 3 and 4, BA-LoRA consistently outperforms LoRA across all tasks, underscoring its superior generalization capabilities. Specifically, BA-LoRA achieves average performance improvements of 2.03% and 2.47% for BERT-L and GPT-2-XL, respectively, on the GLUE benchmark. Similarly, on GLUE-x, BA-LoRA surpasses LoRA by 1.61% and 1.90% for BERT-L and GPT-2-XL, respectively. These results substantiate the effectiveness of our proposed regularization terms in mitigating the negative impacts of noise in pre-training and enhancing model robustness.

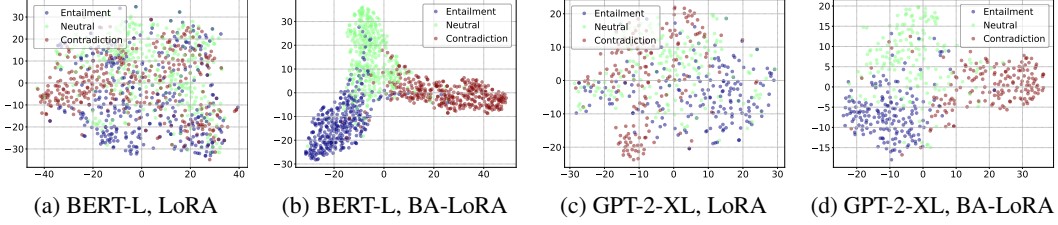

| (a) BERT-L, LoRA | (b) BERT-L, BA-LoRA | (c) GPT-2-XL, LoRA | (d) GPT-2-XL, BA-LoRA |

Figure 1: t-SNE Visualizations Comparing Last Hidden Layer Features of BERT-L and GPT-2-XL Fine-Tuned with LoRA and BA-LoRA on a Mini Subset of the GLUE Dataset.

### 4.3.3 ANALYSIS ON MITIGATING IMBALANCED DATA

This experiment evaluates the effectiveness of BA-LoRA in addressing imbalanced data. Specifically, using the MNLI dataset, LoRA and BA-LoRA are applied to fine-tune the BERT-L and GPT-2-XL models, respectively. The hidden layer features of the last training step are extracted and visualized using t-SNE (Van der Maaten & Hinton, 2008) technology for comparison.

As shown in Figure 1, the models fine-tuned with standard LoRA in sub-figures (a) and (c) have low discrimination between categories and obvious category mixing. In contrast, the models fine-tuned with BA-LoRA in sub-figures (b) and (d) have clearer category separation, especially the results of BERT-L, which have higher intra-category clustering and clearer boundaries. These analyses show that BA-LoRA can effectively alleviate the impact of imbalanced data in pre-training.

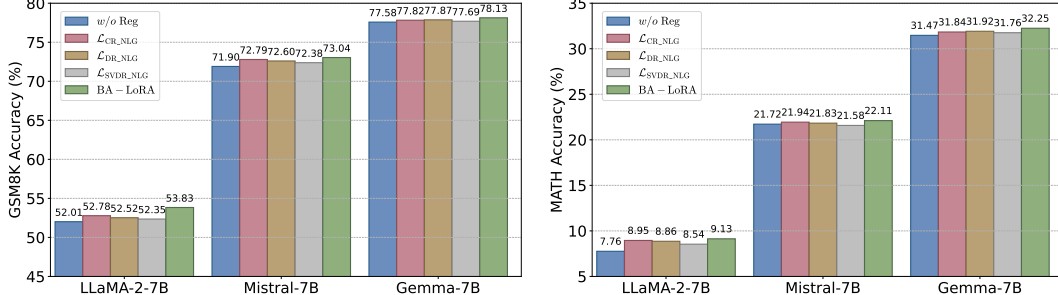

Figure 2: Ablation study results of BA-LoRA regularizations on the GSM8K and MATH datasets. Here, "Reg" stands for "Regularization" and "$w/o$ Reg" means "without regularization". $\mathcal{L}_{\text{CR\_NLG}}$, $\mathcal{L}_{\text{DR\_NLG}}$, and $\mathcal{L}_{\text{SVDR\_NLG}}$ denote the application of only the corresponding regularization, and "BA-LoRA" refers to the baseline using all regularizations.

Table 5: Ablation results of BA-LoRA regularization on NLU tasks. Here, "Reg" stands for "regularization", and "$w/o$ Reg" means "no regularization". $\mathcal{L}_{\text{CR\_NLU}}$, $\mathcal{L}_{\text{DR\_NLU}}$, and $\mathcal{L}_{\text{SVDR\_NLU}}$ mean that only the corresponding regularization is applied, and "BA-LoRA" refers to the baseline using all regularization. The best and second-best results are highlighted in **bold** and underline.

| Method | MNLI | SST-2 | MRPC | CoLA | QNLI | QQP | RTE | SST-B | Avg |
|--------|------|-------|------|------|------|-----|-----|-------|-----|
| $w/o$ Reg | 90.47 | 95.81 | 91.48 | 72.27 | 94.41 | 92.21 | 87.14 | 91.93 | 89.47 |
| $\mathcal{L}_{\text{CR\_NLU}}$ | 90.84 | 96.27 | 91.65 | 72.68 | 94.64 | 92.47 | 87.59 | 92.11 | 89.78 |
| $\mathcal{L}_{\text{DR\_NLU}}$ | 90.77 | 96.09 | 91.81 | 72.44 | 94.44 | 92.41 | 87.37 | 91.89 | 89.65 |
| $\mathcal{L}_{\text{SVDR\_NLU}}$ | 90.63 | 95.96 | 91.37 | 72.37 | 94.58 | 92.39 | 87.35 | 92.08 | 89.59 |
| BA-LoRA | **90.92** | **96.25** | **91.83** | **72.79** | **94.84** | **92.59** | **87.87** | **92.15** | **89.91** |

### 4.3.4 ABLATION STUDY

This ablation experiment aims to analyze the impact of three regularization terms ($\mathcal{L}_{\text{CR\_NLG}}$, $\mathcal{L}_{\text{DR\_NLG}}$, $\mathcal{L}_{\text{SVDR\_NLG}}$) in BA-LoRA on model performance. The experiment selected three models, LLaMA-2-7B, Mistral-7B, and Gemma-7B, and evaluated them on GSM8K and MATH datasets. To ensure clear results, we only tested single regularization terms to reveal their independent contributions.

As shown in Figure 2, the model without regularization ("$w/o$ Reg") produced the lowest performance across both datasets. In contrast, introducing different regularization terms led to varying degrees of performance improvement. Specifically, $\mathcal{L}_{\text{CR\_NLG}}$ contributes to the equilibrium gain for both datasets. The effect of $\mathcal{L}_{\text{DR\_NLG}}$ on Gemma-7B is significantly stronger relative to $\mathcal{L}_{\text{CR\_NLG}}$. Ultimately, the model incorporating all three regularization terms achieved the highest performance on both datasets. These findings validate the regularization strategy and show that combining the terms in BA-LoRA further enhances model generalization. These findings confirm the effectiveness of the proposed regularization strategy, and further indicate that combining the regularization terms in BA-LoRA enhances the model's generalization ability.

To further assess the impact of our proposed regularization terms on model performance, we conducted an ablation study focusing on three terms designed for NLU tasks in BA-LoRA: $\mathcal{L}_{\text{CR\_NLU}}$, $\mathcal{L}_{\text{DR\_NLU}}$, and $\mathcal{L}_{\text{SVDR\_NLU}}$. We employed the DeBERTa-v3-base model and evaluated it across several tasks from the GLUE benchmark.

As shown in Table 5, the model without any regularization (denoted as "$w/o$ Reg") exhibited the lowest performance, achieving an average score of 89.47. Introducing each regularization term individually led to performance improvements across various tasks. Notably, $\mathcal{L}_{\text{CR\_NLU}}$ consistently provided substantial gains, achieving an average score of 89.78, second only to the full BA-LoRA model. The term $\mathcal{L}_{\text{DR\_NLU}}$ showed particularly strong performance on the MRPC task, while $\mathcal{L}_{\text{SVDR\_NLU}}$ delivered balanced improvements across most tasks. Finally, BA-LoRA, incorporating all three regularization terms, achieved the highest overall performance with an average score of 89.91.

These results demonstrate that each regularization term contributes to enhancing model performance, and their combination in BA-LoRA provides optimal generalization on NLU tasks.

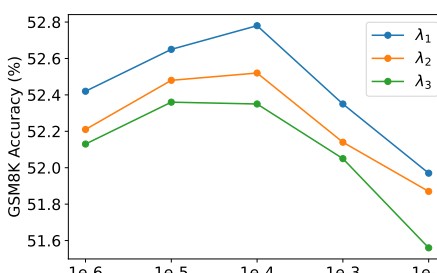 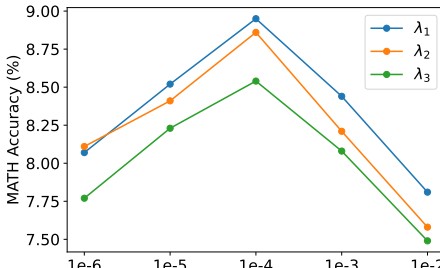

Figure 3: Impact of Different Regularization Hyperparameter Sizes for $\mathcal{L}_{\text{task\_NLG}}$ on Fine-Tuned Model Performance, with Each Line Representing Results Using a Single Hyperparameter Value (Without Combinations).

### 4.3.5 HYPERPARAMETER ANALYSIS

This experiment aims to explore the impact of different sizes of regularization term hyperparameters on the performance of fine-tuned models for NLG tasks. The three regularization hyperparameters tested in the experiment are $\lambda_1$, $\lambda_2$, and $\lambda_3$, with values selected from the range of $1e-6$, $1e-5$, $1e-4$, $1e-3$, and $1e-2$. We first fine-tune the LLaMA-2-7B model on the MetaMathQA dataset and then evaluate its performance on the GSM8K and MATH datasets.

As illustrated in Figure 3, the model achieves optimal performance on both datasets when $\lambda_1$, $\lambda_2$, and $\lambda_3$ are set to $1e^{-4}$. For smaller regularization values (e.g., $1e^{-6}$ or $1e^{-5}$), the regularization effect is minimal, leading to limited performance gains. In contrast, with larger regularization values (e.g., $1e^{-3}$ or $1e^{-2}$), the model's performance decreases significantly. This decline occurs because higher regularization values impose stronger constraints on the training process, limiting the model's capacity to learn and resulting in performance degradation.

## 5 CONCLUSION

This paper introduces Bias-Alleviating Low-rank Adaptation (BA-LoRA), a novel parameter-efficient fine-tuning method designed to mitigate catastrophic inheritance in pre-trained language models. BA-LoRA incorporates three key components: consistency regularization, diversity regularization, and singular value decomposition regularization. These components work in concert to preserve pre-training knowledge, enhance output diversity, and improve model generalization. Extensive experiments demonstrate that BA-LoRA consistently outperforms existing baselines on various NLG and NLU tasks while robust to noisy and imbalanced pre-training data. Furthermore, our ablation studies confirm the effectiveness of the three regularization terms both individually and in combination. These results highlight the potential of BA-LoRA as a general-purpose fine-tuning method for pre-trained language models and effectively address the key challenges of deploying these models in real applications.

## 6 ETHICS STATEMENT

This study aims to develop and evaluate BA-LoRA, a novel parameter-efficient fine-tuning method designed to mitigate bias and enhance the performance of LLMs. Our research utilizes existing open-source public datasets for both fine-tuning and evaluation purposes. For Natural Language

Generation tasks, we employed widely recognized datasets within the research community, including MetaMathQA, CodeFeedback, and WizardLM-Evol-Instruct. These datasets have no known ethical concerns. For Natural Language Understanding tasks, we utilized the GLUE and GLUE-X benchmarks, standard evaluation datasets in machine learning. We are committed to the responsible development and application of AI technologies. Throughout this research, we will continue to monitor and address any ethical issues that may arise.

## 7 REPRODUCIBILITY

To ensure the reproducibility of our results, we provide a detailed description of our experimental setup in Section 4.2 and Appendix Section B, including model introduction, dataset introduction, hyperparameter configuration, and evaluation procedures. All models and datasets used are publicly available. In addition, we have refined the implementation scripts and fine-tuning strategies to facilitate independent verification. Our source code and pre-trained model weights will be made public upon acceptance of this paper, ensuring that our results are fully transparent and reproducible.

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

# Appendix

CONTENTS

## A  BACKGROUND

### A.1  CHALLENGES OF BIAS AND NOISE IN PRE-TRAINING DATA

Bias and noise within pre-training datasets present significant hurdles in constructing dependable machine-learning models. Mislabeled data and imbalanced distributions can lead to models that not only underperform on downstream tasks but also reinforce existing biases in the data (Torralba & Efros, 2011; Barocas & Selbst, 2016; Mehrabi et al., 2021). This issue is especially problematic in large-scale datasets, where manual curation is impractical. Consequently, reliance on automated data collection methods may introduce various inaccuracies and biases (Northcutt et al., 2021; Birhane & Prabhu, 2021). The challenge becomes even more severe when dealing with real-world, instance-dependent label noise. Models trained on such data may inadvertently learn these inaccuracies, resulting in poor generalization (Frénay & Verleysen, 2013; Song et al., 2022; Algan & Ulusoy, 2021). Addressing these challenges is essential for advancing machine learning and ensuring models are both effective and equitable.

## A.2 MITIGATING BIAS AND NOISE THROUGH PARAMETER-EFFICIENT FINE-TUNING METHODS

To counteract the adverse effects of bias and noise in pre-training data, parameter-efficient fine-tuning methods have emerged as promising solutions. These approaches aim to adapt pre-trained models to new tasks with minimal parameter updates, thereby reducing the risk of overfitting to noisy or biased data (Houlsby et al., 2019a; Zaken et al., 2021; Lester et al., 2021). Techniques such as integrating lightweight adaptation modules (Pfeiffer et al., 2020; Jang et al., 2021), utilizing prefix tuning (Li & Liang, 2021b; Liu et al., 2021), and employing low-rank adaptations (Hu et al., 2021; Ding et al., 2023) enable efficient model refinement while preserving the valuable representations acquired during pre-training. Selectively fine-tuning specific model components can enhance performance on downstream tasks, improve generalization, and reduce the influence of noise and bias (Zaken et al., 2021; Mahabadi et al., 2021; Guo et al., 2020). This strategy not only results in more robust models but also contributes to the development of fairer AI systems by directly addressing fundamental data quality issues.

## A.3 EXAMPLES OF NOISE IN PRE-TRAINING DATA

Pre-training data typically originates from large-scale internet sources, which inevitably contain noise and imbalance. Many advanced pre-trained models, such as LLaMA-2-7B/13B (Touvron et al., 2023), Mistral-7B-v0.1 (Jiang et al., 2023), Gemma-7B (Jiang et al., 2023), and GPT-4 (OpenAI, 2023), are trained using large amounts of unlabeled internet text data. These datasets are often not thoroughly cleaned or corrected, leading to training corpora that include irrelevant or inaccurate information. Consequently, during the subsequent fine-tuning phase, models struggle to effectively filter out these undesirable contents, adversely affecting their performance on downstream tasks.

Given the noise and imbalance issues present in pre-training data, understanding the specific types of noise is crucial for improving model performance. Below, we summarize some common examples of noise found in pre-training datasets:

- **Low quality bias:**

    *Duplicity:* The presence of identical or similar content in the data can lead to overfitting and privacy leakage risks (Elazar et al., 2023), (Carlini et al., 2022), (Hernandez et al., 2022).

    *Corruption/noise:* Inconsistent or erroneous inputs in the training data can affect model robustness and downstream task performance (Elazar et al., 2023), (Fan et al., 2024), (Caswell et al., 2021).

    *Contamination:* Leakage of the test set into the training set may lead to distorted model evaluation results (Roberts et al., 2023), (Schaeffer, 2023), (Jiang et al., 2024b).

- **Skewed distribution bias:**

    *Category imbalance:* Too few samples of certain categories cause the model to perform poorly in predicting these categories, producing bias (Xu et al., 2023a), (Zhu et al., 2024), (Parashar et al., 2024).

- **Unethical content bias:**

    *Toxic and harmful content:* The training data may contain offensive, biased, or harmful content that may cause the model to generate harmful or inappropriate outputs (Zou et al., 2023), (Sun et al., 2024).

## B DETAILS OF MODELS AND DATASETS

## B.1 DETAILS OF MODELS

We use a variety of pre-trained language models, including Meta AI's LLaMA-2-7B and LLaMA-2-13B (Touvron et al., 2023) and the latest LLaMA-3-8B and LLaMA-3-70B (AI@Meta, 2024), which have good performance in natural language generation tasks. In addition, we also use Mistral

AI's Mistral-7B-v0.1 (Jiang et al., 2023) optimized for medium-sized model efficiency, and Google's lightweight open-source model Gemma-7B (Jiang et al., 2023), which performs well in tasks such as question-answering summarization and reasoning. Alibaba Cloud's Qwen-1.5-7B (Bai et al., 2023) model also provides strong language understanding and generation capabilities, while the 34B parameter Yi-1.5-34B (Young et al., 2024) is designed for high-level language tasks. DeepSeek-MoE-16B (Dai et al., 2024) is a model that uses expert routing to increase capacity without significantly increasing computational costs. Mixtral-8x7B-v0.1 (Jiang et al., 2024a) is a Sparse Mixture of Expert models that efficiently utilizes active parameters to outperform larger models like Llama 2 70B and GPT-3.5 across several benchmarks. We also leveraged mature models such as BERT-Large (Devlin et al., 2018), RoBERTa-large (Liu et al., 2019), DeBERTa-v3-base (He et al., 2021b), and GPT-2-XL (Radford et al., 2019), which continue to set standards in natural language processing and text generation tasks.

Table 6: Comparison of Pre-trained Data and Methods for Various Language Models.

| Model | Pre-trained Data | Pre-training Method |
|---|---|---|
| BERT-L (Devlin et al., 2018) | BooksCorpus and English Wikipedia | Masked Language Modeling |
| RoBERTa-L (Liu et al., 2019) | BooksCorpus and English Wikipedia | Masked Language Modeling |
| GPT-2-XL (Radford et al., 2019) | WebText | Autoregressive Language Modeling |
| DeBERTa-v3-base (He et al., 2021b) | Wikipedia, BooksCorpus, OpenWebText, CC-News, and Stories | Replaced Token Detection with GDES |

Table 6 presents an overview of the pre-trained language models used in our study. BERT-Large (BERT-L) and RoBERTa-Large (RoBERTa-L) are pre-trained on the BooksCorpus and English Wikipedia datasets using a masked language modeling objective. In contrast, GPT-2-XL is pre-trained on WebText with an autoregressive language modeling objective. Additionally, DeBERTa-v3-base is trained on a diverse dataset comprising Wikipedia, BooksCorpus, OpenWebText, CC-News, and Stories, utilizing a replaced token detection objective with Gradient Disentangled Embedding Sharing (GDES). These models span a variety of architectures and pre-training strategies, offering a robust basis for evaluating the performance of our proposed approach.

## B.2 DETAILS OF DATASETS

Table 7 provides an overview of the GLUE benchmark datasets and their evaluation metrics. The GLUE benchmark comprises a diverse set of natural language understanding tasks, including grammatical acceptability (CoLA), sentiment analysis (SST-2), paraphrase detection (MRPC and QQP), sentence similarity (STS-B), natural language inference (MNLI, QNLI, and RTE), and coreference resolution (WNLI). The number of training examples varies significantly across datasets, from as few as 634 in WNLI to as many as 393,000 in MNLI. Tasks involve binary or multi-class classification, with up to five classes in STS-B. Evaluation metrics are tailored to each task, employing accuracy, F1 score, Matthews correlation coefficient, and Pearson/Spearman correlation coefficients where appropriate. This comprehensive suite serves as a standard benchmark for assessing and comparing the performance of models across a wide array of linguistic challenges.

Table 8 summarizes the GLUE-X out-of-domain tasks employed for evaluating transfer performance. The datasets cover a broad spectrum of natural language understanding tasks, including natural language inference (SNLI, HANs, SciTail, MNLI mismatched), sentiment analysis (IMDB), question answering (NewsQA), semantic relatedness (SICK), and grammatical error detection (Grammar Test). Each task involves binary classification, with test sizes ranging from 9,832 samples (MNLI mismatched) to 570,152 samples (SNLI). Accuracy is the primary evaluation metric across most datasets, except for the Grammar Test, which uses the Matthews correlation coefficient. These diverse tasks provide a comprehensive benchmark for assessing the models' ability to generalize across different domains and tasks.

Table 9 summarizes the evaluation metrics for the natural language generation (NLG) tasks. Specifically, we use Accuracy for GSM8K and MATH; Pass@1 for HumanEval and MBPP, indicating the percentage of first generated code snippets that pass all unit tests; and GPT-4 Evaluation for MT-Bench, where GPT-4 assesses the quality of the model's responses.

Table 7: GLUE Benchmark Datasets and Evaluation Metrics

| Dataset | Task Type | Classes | Train Examples | Metric | Description |
|---------|-----------|---------|----------------|--------|-------------|
| CoLA | Acceptability | 2 | 8.5k | Matthews Corr. | Grammatical acceptability |
| SST-2 | Sentiment | 2 | 67k | Accuracy | Sentiment analysis |
| MRPC | Paraphrase | 2 | 3.7k | Accuracy/F1 | Paraphrase detection |
| QQP | Paraphrase | 2 | 364k | Accuracy/F1 | Duplicate question detection |
| STS-B | Similarity | 5 | 7k | Pearson/Spearman Corr. | Sentence similarity |
| MNLI | NLI | 3 | 393k | Accuracy | Multi-genre NLI |
| QNLI | NLI/QA | 2 | 108k | Accuracy | QA/NLI converted from SQuAD |
| RTE | NLI | 2 | 2.5k | Accuracy | Textual entailment |
| WNLI | Coreference | 2 | 634 | Accuracy | Winograd Schema Challenge |

Table 8: Summary of GLUE-X Out-of-Domain Tasks for Transfer Performance Evaluation

| Dataset | Task Type | Classes | Train Examples | Metric | Description |
|---------|-----------|---------|----------------|--------|-------------|
| SNLI | NLI | 2 | 570k | Accuracy | Sentence-level inference tasks |
| IMDB | Sentiment | 2 | 50k | Accuracy | Movie review sentiment analysis |
| HANs | NLI | 2 | 60k | Accuracy | Adversarial NLI examples to test models |
| NewsQA | QA | 2 | 119k | Accuracy | QA from news articles |
| SICK | Semantic Relatedness | 2 | 9.8k | Accuracy | Semantic relatedness and entailment |
| Grammar Test | Grammar Detection | 2 | 304k | Matthews Corr. | Grammatical error detection |
| SciTail | NLI | 2 | 26.5k | Accuracy | Science question entailment |
| MNLI mismatched | NLI | 2 | 9.8k | Accuracy | NLI with mismatched genres |

Table 9: Evaluation Metrics for NLG Datasets

| Datasets | GSM8K | MATH | HumanEval | MBPP | MT-Bench |
|----------|-------|------|-----------|------|----------|
| Metric | Accuracy | Accuracy | Pass@1 | Pass@1 | GPT-4 Evaluation |

### B.3 SPECIFIC HYPERPARAMETER SETTINGS OF ROBERTA-LARGE AND DEBERTA-V3-BASE ON GLUE

We fine-tuned the RoBERTa-large and DeBERTa-v3-base models on the GLUE benchmark datasets using carefully selected hyperparameters tailored to each task. For RoBERTa-large, we trained on MNLI and SST-2 for 10 epochs with a batch size of 32; MNLI employed a learning rate of $1 \times 10^{-4}$, while SST-2 used $2 \times 10^{-4}$, both with `LoRA_alpha` set to 16. Smaller datasets such as MRPC, CoLA, and RTE were trained for 20 epochs with batch sizes of 16, utilizing higher learning rates ranging from $3 \times 10^{-4}$ to $6 \times 10^{-4}$ and `LoRA_alpha` values of 8 or 16. For DeBERTa-v3-base, MNLI was trained for 5 epochs with a batch size of 16, a learning rate of $5 \times 10^{-5}$, and `LoRA_alpha` set to 8. Datasets such as SST-2 and MRPC were trained for 20 epochs with batch sizes of 16 or 32, learning rates between $3 \times 10^{-5}$ and $2 \times 10^{-4}$, and `LoRA_alpha` of 8. Notably, RTE was trained for 50 epochs with a batch size of 16, a learning rate of $1 \times 10^{-4}$, and `LoRA_alpha` of 8. The `LoRA_alpha` parameter was set to either 8 or 16, depending on the model and dataset. In all cases, the `LoRA_rank` was set to 8. These hyperparameters were meticulously chosen to suit the specific requirements of each dataset, ensuring rigorous and optimal training across tasks such as natural language inference, sentiment analysis, paraphrase detection, linguistic acceptability, and semantic textual similarity.

### B.4 SPECIFIC HYPERPARAMETER SETTINGS OF BERT-L AND GPT-2-XL ON GLUE AND GLUE-X

To ensure consistent and reliable performance, the BERT-Large (BERT-L) and GPT-2-XL models were trained on the GLUE benchmark tasks using three different random seeds per task over 10 epochs. A hyperparameter search was conducted over learning rates $\{2 \times 10^{-5}, 3 \times 10^{-5}, 5 \times 10^{-5}\}$, and a batch size of 32 was chosen to balance computational efficiency and memory usage. For fine-tuning, the training schedule was adjusted to 20 epochs for smaller datasets, while larger datasets such as QNLI, MNLI, and QQP were trained for 5 epochs. Learning rates were explored within $\{2 \times 10^{-4}, 3 \times 10^{-4}, 5 \times 10^{-4}\}$. The parameters were set with `LoRA_rank` $= 8$ and `LoRA_alpha` $= 16$,

with the batch size reduced to 16 due to increased model complexity. All other parameters, including `max_length`, adhered to Hugging Face Transformers guidelines[1].

Regarding the GLUE-x tasks, BERT-L and GPT-2-XL models trained on GLUE were evaluated without further fine-tuning. GLUE-x encompasses 13 out-of-distribution (OOD) tasks, introducing domain shifts. For sentiment analysis, models fine-tuned on SST-2 were evaluated on test sets from IMDB (Maas et al., 2011), Yelp (Zhang et al., 2015), Amazon (Kaushik et al., 2019), and Flipkart (Vaghani & Thummar, 2023), offering a broader assessment of domain variability and testing the robustness beyond SST-2.

For t-SNE visualization, we used the MNLI subset from GLUE due to its diverse linguistic styles and label distributions. Training was limited to one epoch to expedite the process, while still providing insights into how well the models differentiate between classes and sentence structures.

### B.5 MODEL EVALUATION DETAILS

For evaluation, we employed publicly available frameworks. The model's code generation capabilities were assessed using datasets like HumanEval and MBPP through the BigCode Evaluation Harness[2]. Instruction-following performance was evaluated using MTBench[3].

## C MORE EXPERIMENTS

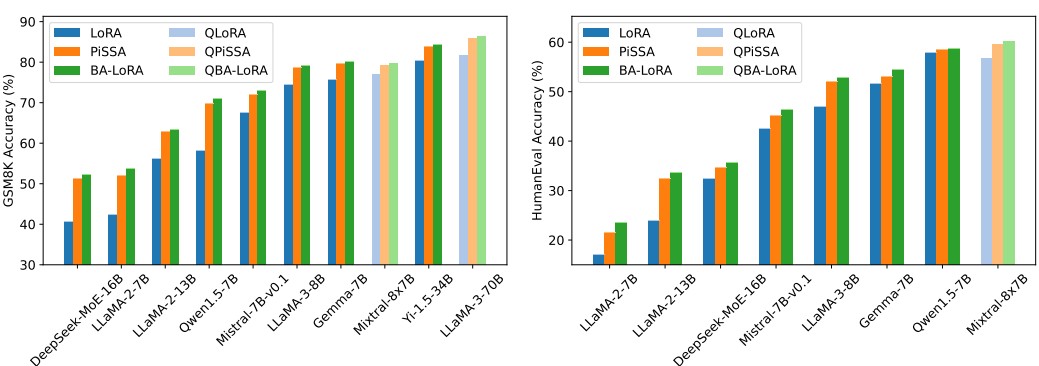

Figure 4: Performance Comparison of Different Models on GSM8K and HumanEval Benchmarks.

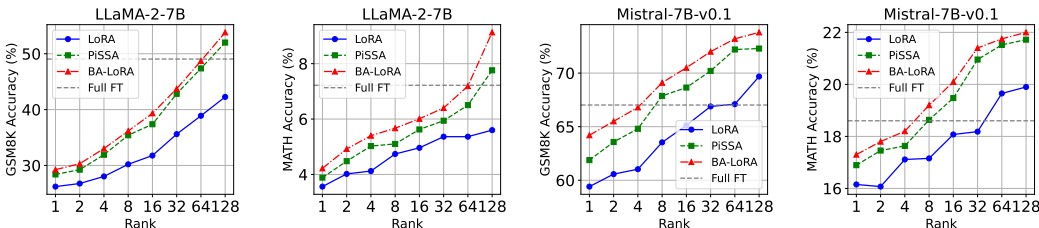

Figure 5: Performance Comparison of LoRA and BA-LoRA Across Various Ranks.

### C.1 ANALYSIS ON DIFFERENT SIZES AND TYPES OF MODELS

This experiment compares LoRA, PiSSA, and BA-LoRA across ten models: LLaMA-2-7/13B (Touvron et al., 2023), LLaMA-3-8B/70B (AI@Meta, 2024), Mistral-7B-v0.1 (Jiang et al., 2023),

---

[1] https://github.com/huggingface/transformers
[2] https://github.com/bigcode-project/bigcode-evaluation-harness
[3] https://github.com/lm-sys/FastChat

Gemma-7B (Jiang et al., 2023), Qwen1.5-7B (Bai et al., 2023), Yi-1.5-34B (Young et al., 2024) and Mixture-of-Experts (MoE (Shazeer et al., 2017)) models: DeepSeek-MoE-16B (Dai et al., 2024) and Mixtral-8x7B-v0.1 (Jiang et al., 2024a). These models were fine-tuned on the MetaMathQA-100K and CodeFeedback-100K datasets and evaluated on the GSM8K and HumanEval benchmarks. As depicted in Figure 4, BA-LoRA consistently surpasses LoRA and PiSSA across all models and tasks, underscoring its superior ability to enhance model generalization.

## C.2 EVALUATING THE PERFORMANCE OF DIFFERENT RANKS

We compare the performance of BA-LoRA, LoRA, and PiSSA at different ranks using LLaMA-2-7B and Mistral-7B-v0.1 models. Each method is fine-tuned for one epoch on the MetaMathQA-100K dataset with ranks ranging from 1 to 128 and evaluated on the GSM8K and MATH datasets. As shown in Figure 5, BA-LoRA consistently outperforms LoRA and PiSSA across all rank settings and datasets. As the rank increases, the performance of BA-LoRA and PiSSA surpasses full parameter fine-tuning. However, BA-LoRA performs better, especially on Mistral-7B-v0.1.

## C.3 IMPACT OF REGULARIZATION ON LoRA VARIANTS PERFORMANCE

Table 10: Effect of regularization term in different LoRA variants. The best results are highlighted in **bold**.

| Methods | #Params | GSM8K | MATH | HumanEval | MBPP | MT-Bench | Avg |
|---------|---------|-------|------|-----------|------|----------|-----|
| Full FT | 6738M | 49.05 | 7.22 | 21.34 | 35.59 | 4.91 | 23.62 |
| LoRA | 320M | 42.47 | 5.60 | 17.03 | 31.48 | 4.62 | 20.24 |
| LoRA + Reg | 320M | 49.26 | 6.84 | 20.16 | 32.75 | 4.75 | 22.75 |
| DoRA | 321M | 42.12 | 6.28 | 16.97 | 22.07 | 4.53 | 18.46 |
| PiSSA | 320M | 52.01 | 7.76 | 21.55 | 33.09 | 4.87 | 23.86 |
| DoRA + Reg | 321M | 52.80 | 8.05 | 21.94 | 34.61 | 5.02 | 24.48 |
| BA-LoRA | 320M | **53.83** | **9.13** | **23.58** | **36.86** | **5.11** | **25.70** |

In this experiment, we evaluated the impact of the regularization terms on multiple LoRA variants using the LLaMA-2-7B. Table 10 shows the performance comparison of LoRA, DoRA, PiSSA, and BA-LoRA with and without regularization terms, where "Reg" refers to the three regularization terms designed for each NLG task.

The experimental results indicate that incorporating regularization terms into both LoRA and DoRA architectures significantly enhances their performance across all evaluated tasks. This finding demonstrates that regularization techniques are broadly effective when applied to different LoRA variants. Furthermore, BA-LoRA, which integrates PiSSA with regularization, achieves the best performance across various tasks and substantially improves the model's generalization capabilities.

## C.4 t-SNE VISUALIZATIONS OF FEATURE EVOLUTION DURING THE FINE-TUNING WITH LoRA AND BA-LoRA

This section provides more detailed t-SNE visualization results to compare the feature evolution during fine-tuning of LoRA and BA-LoRA.

Figure 6 shows that during LoRA fine-tuning of BERT-L, the feature separation is slow, with the class distributions remaining scattered and overlapping even towards the end. In contrast, Figure 7 demonstrates that with BA-LoRA fine-tuning, class separation begins earlier and is much clearer, ultimately forming a distinct "Y" shape with well-defined class boundaries.

Similarly, Figure 8 shows that during LoRA fine-tuning of GPT-2 XL, the feature clusters remain scattered and overlapping throughout the training, with only minimal separation between classes by the final steps. In contrast, Figure 9 demonstrates that BA-LoRA fine-tuning results in much clearer and more distinct class separation, with well-defined boundaries emerging earlier in the training process and becoming more pronounced over time.

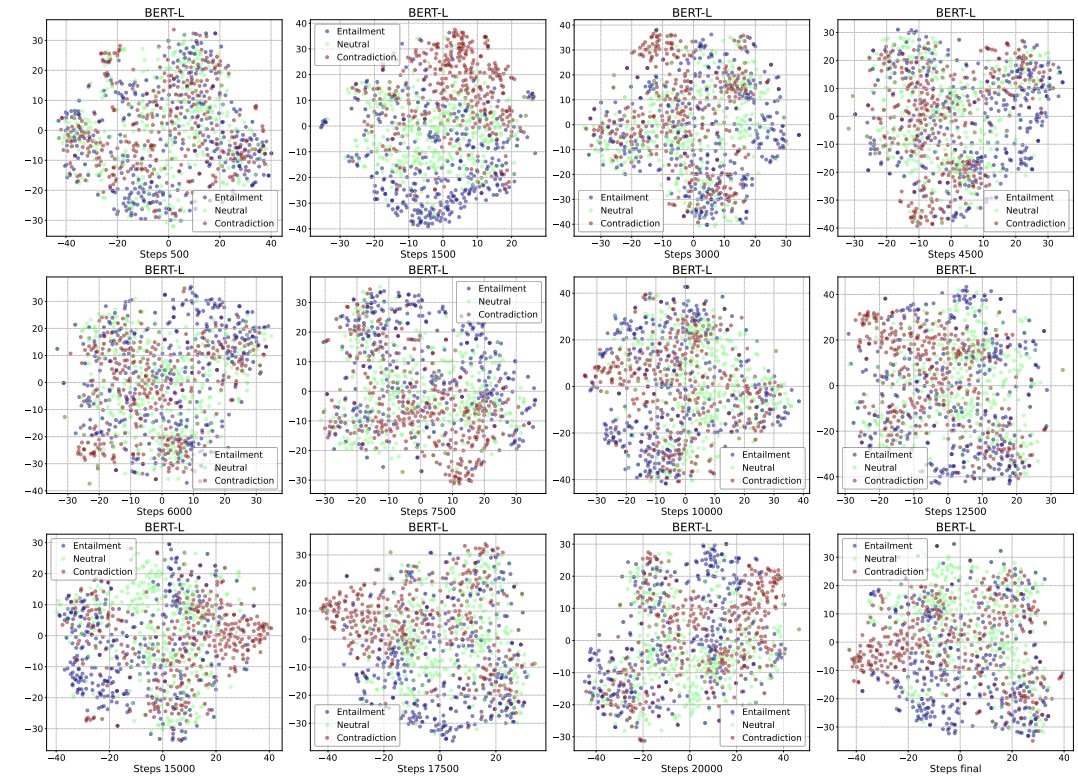

Figure 6: t-SNE Visualization of Feature Evolution during LoRA Fine-Tuning of BERT-L.

# D   MORE DISCUSSIONS

Here, we offer further insights into our work.

## D.1   LIMITATIONS

While BA-LoRA demonstrates significant improvements in mitigating catastrophic inheritance and enhancing model performance, several limitations warrant further investigation. Firstly, our evaluations primarily focus on English language tasks, which may limit the generalizability of our findings to other languages and specialized domains. Additionally, the computational overhead introduced by the consistency, diversity, and SVD regularizers adds complexity to the training process, potentially impacting efficiency. Furthermore, the impact of BA-LoRA on other forms of bias, such as fairness and societal stereotypes, remains unexplored. Lastly, the selection and weighting of regularization terms in BA-LoRA are fixed across different tasks, which may not be optimal for all scenarios.

## D.2   FUTURE WORKS

Future research should extend assessments of BA-LoRA to multilingual settings and specialized domains to ensure broader applicability. Exploring optimization techniques could help reduce the computational overhead introduced by the regularizers, balancing performance gains with efficiency. Investigating the impact of BA-LoRA on other forms of bias, including fairness and societal stereotypes, is crucial for developing more equitable models. Additionally, refining the selection and weighting of regularization terms—possibly through automated or dynamic adjustment methods—could enhance adaptability across different tasks and models. Testing the scalability of BA-LoRA on larger models with hundreds of billions of parameters and exploring its integration with other bias mitigation strategies may yield synergistic effects and further improve model robustness.

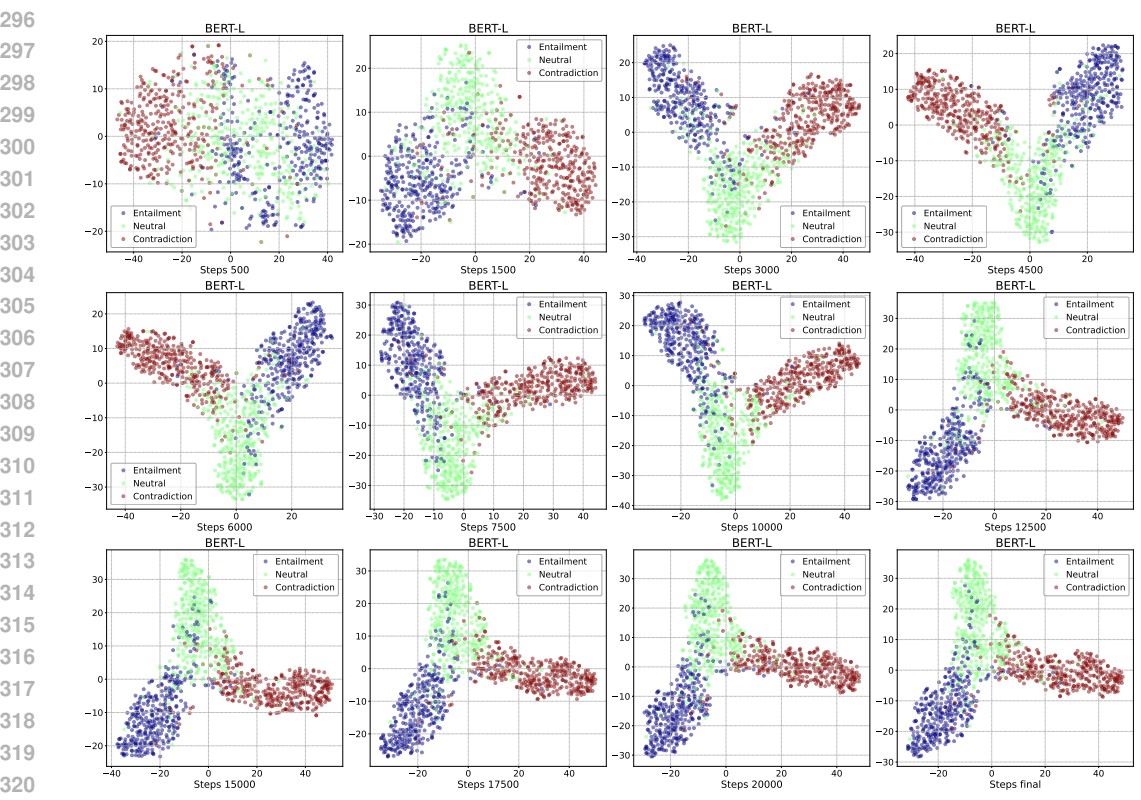

Figure 7: t-SNE Visualization of Feature Evolution during BA-LoRA Fine-Tuning of BERT-L.

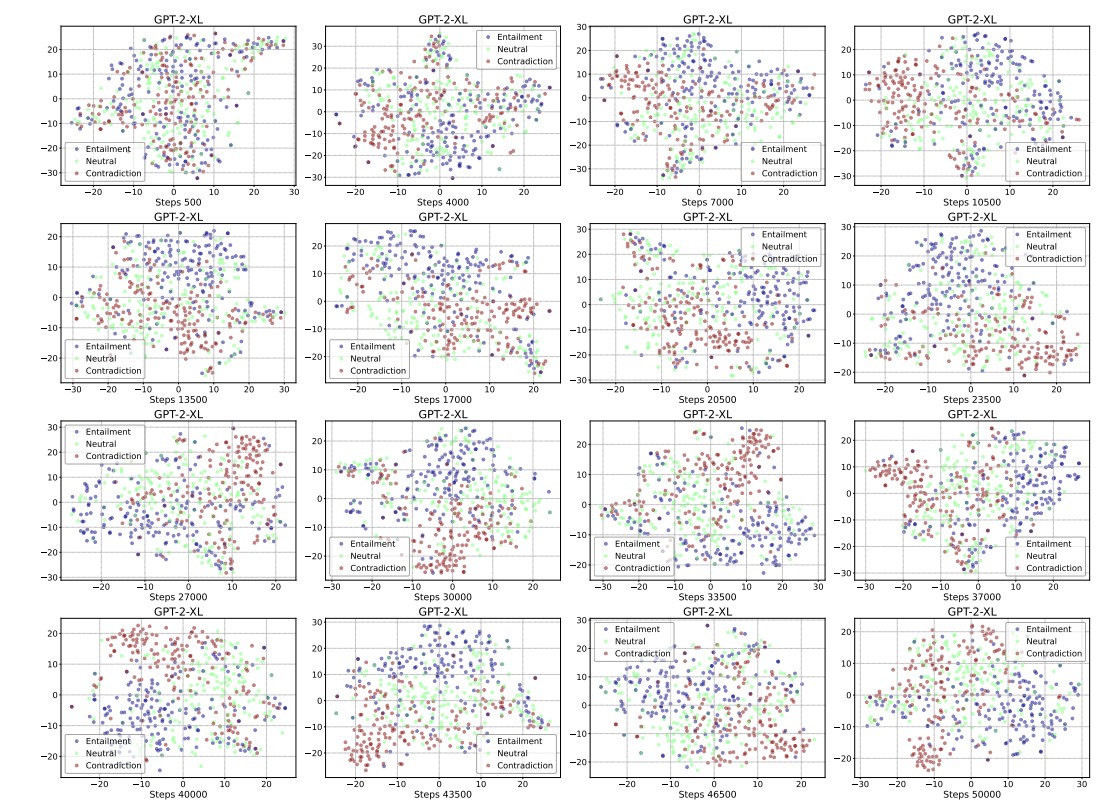

Figure 8: t-SNE Visualization of Feature Evolution during LoRA Fine-Tuning of GPT-2-XL.

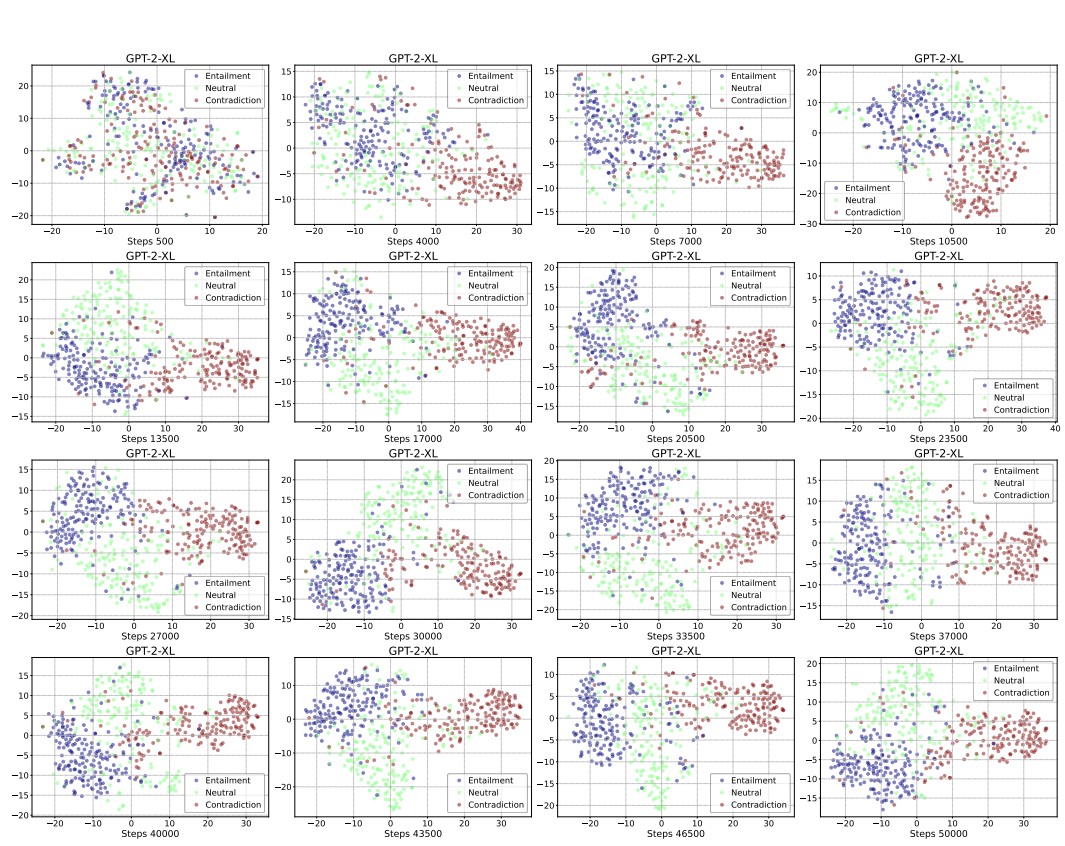

Figure 9: t-SNE Visualization of Feature Evolution during BA-LoRA Fine-Tuning of GPT-2-XL.

