# OpenReview forum: "BA-LoRA: Bias-Alleviating Low-Rank Adaptation to Mitigate Catastrophic Inheritance in Large Language Models"
_ICLR.cc/2025/Conference — Submitted to ICLR 2025_

### Official Review · Reviewer_VRaY · 2024-10-31

**Soundness:** 4
**Presentation:** 3
**Contribution:** 3
**Rating:** 8
**Confidence:** 3

**Summary:**

This paper proposes BA-LoRA, a parameter-efficient fine-tuning method to mitigate the catastrophic inheritance issue in LLMs. Key contributions include (1) three regularization terms for bias mitigation, (2) optimized adaptation methods for NLU/NLG tasks, and (3) comprehensive evaluations across various models and benchmarks.

**Strengths:**

This paper addresses an essential issue for the practical deployment of large language models (LLMs) by proposing BA-LoRA, a parameter-efficient fine-tuning method aimed at mitigating catastrophic inheritance problems.

The motivation is clear and timely, given the growing demand for efficient fine-tuning techniques in LLMs. The work introduces technical innovations, including a novel combination of regularization strategies tailored to handle task-specific characteristics in natural language understanding (NLU) and natural language generation (NLG).

The method effectively integrates with existing LoRA frameworks, enhancing its applicability and relevance. Experimental validation is thorough, covering various models and demonstrating substantial performance improvements over baseline approaches. The analysis is supported by detailed component studies and persuasive qualitative analysis, including t-SNE visualizations, which further illustrate the effectiveness of BA-LoRA.

**Weaknesses:**

However, the paper has some limitations in terms of theoretical analysis, experimental scope, and comparative analysis.

Theoretical analysis is lacking, with no clear basis for the selection of regularization weights and limited discussion on computational complexity and convergence.

The experimental scope is restricted, as the study only tests on English datasets, leaving the method's performance in multilingual contexts unexplored( and also other tasks as well ). Additionally, the analysis of computational overhead is minimal.

**Questions:**

How do you determine the optimal values for the regularization weights (λ1, λ2, λ3)?
What is the computational overhead compared to standard LoRA?
How does the performance fare on non-English datasets or other NLP tasks?
Have you tested the effects on other types of bias beyond those mentioned in the paper?

---

> ### Author Response · Authors · 2024-11-22
> **Response to Reviewer VRaY (Part I)**
>
> > **Q1:** However, the paper has some limitations in terms of theoretical analysis, experimental scope, and comparative analysis.
>
> **A1:** The main task of this paper is to alleviate the catastrophic inheritance problem, and we leave the theoretical study for future work. In addition, we expand the scope of the experiment, adding more evaluation baselines and more comprehensive ablation experiments. We also conduct a more detailed analysis of the computational overhead, including the training time and computational cost during the training process.
>
> > **Q2:** Theoretical analysis is lacking, with no clear basis for the selection of regularization weights and limited discussion on computational complexity and convergence.
>
> **A2:** In the revised version, we have provided a detailed explanation of this choice. The goal of selecting regularization weights is to balance the model's generalization ability with the regularization effect during training. We conducted hyperparameter optimization on the validation set to ensure that the selected regularization weights effectively improve model performance while avoiding excessive regularization. Additionally, we thoroughly discuss the impact of regularization weights on computational complexity and convergence, and we validate their appropriateness through our experiments.
>
> > **Q3:** The experimental scope is restricted, as the study only tests on English datasets, leaving the method's performance in multilingual contexts unexplored( and also other tasks as well ). Additionally, the analysis of computational overhead is minimal.
>
> **A3:** The core goal of our research is to alleviate the catastrophic inheritance problem in pre-trained models, which mainly stems from the selection of pre-training data. Currently, pre-training datasets are mainly English data, so our experiments focus on the English dataset for Supervised Fine-Tuning (SFT). We plan to explore the research of multilingual datasets in future work.
>
> In the analysis of computational overhead, we supplemented the detailed comparison between BA-LoRA and standard LoRA, covering training time and computational load.
>
> > **Q4:** How do you determine the optimal values for the regularization weights (λ1, λ2, λ3)?
>
> **A4:** The main goal of choosing appropriate regularization hyperparameters is to balance the generalization ability of the model and the regularization effect during training. We mainly followed the following steps when choosing regularization weights:
>
> 1. **Split validation set**: Split a part of the training data as a validation set to ensure that it does not participate in the training process and is specifically used to evaluate the performance of the model.
>
> 2. **Hyperparameter adjustment**: Within a reasonable range, we adjust the regularization weights by using hyperparameter optimization algorithms (such as random search) and evaluate the performance of the model on the validation set under different settings. The reasonable range is usually set based on previous experience or pre-experimental results.
>
> 3. **Select the best parameters**: Based on the performance indicators on the validation set, the regularization weight combination that can best improve the generalization ability of the model is selected.

---

> > ### Author Response · Authors · 2024-11-22
> > **Response to Reviewer VRaY (Part II)**
> >
> > > **Q5:** What is the computational overhead compared to standard LoRA?
> >
> > **A5:** First, the number of fine-tuning parameters of BA-LoRA and standard LoRA is consistent. For example, when using the LLaMA-2-7B model for NLG tasks, the number of fine-tuning parameters of LoRA and BA-LoRA is 320M; when using the RoBERTa-large model for NLU tasks, the number of fine-tuning parameters of both is 1.84M. This shows that the computational resource consumption of BA-LoRA is comparable to that of standard LoRA.
> >
> > Second, the introduction of three regularization terms does bring a certain amount of computational overhead, but we optimized these regularization terms when designing them to ensure that they improve model performance while keeping the impact on computational overhead within an acceptable range. To further evaluate the computational overhead, we supplemented the detailed comparison of BA-LoRA and standard LoRA, covering aspects such as training time, memory usage, and computational load.
> >
> > > **Q6:** How does the performance fare on non-English datasets or other NLP tasks?
> >
> > **A6:** The core goal of our research is to alleviate the catastrophic inheritance problem in pre-trained models, which mainly stems from the selection of pre-training data. Currently, pre-training datasets are mainly English data, so our experiments focus on the English SFT dataset. We plan to explore the research of multilingual datasets in future work.
> >
> > In order to add other NLP tasks, we have added more comprehensive baseline comparison experiments in the revised version. In addition, we have also added ablation analysis of regularization terms related to NLG tasks.
> >
> > > **Q7:** Have you tested the effects on other types of bias beyond those mentioned in the paper?
> >
> > **A7:** In this study, we focus on the disastrous consequences of bias and imbalanced data. In the revised version of the paper, we will discuss more different types of bias, mainly including the following categories:
> >
> > 1\. **Low-quality bias**:
> >
> > - **Repetition**: Including repeated or semantically similar content may cause the model to overfit and memorize, thereby increasing privacy risks [1], [2], [3].
> >
> > - **Noise/corruption**: Including unnatural or mismatched training data may reduce the generalization ability of the model and affect the performance of downstream tasks [1], [4], [5].
> >
> > - **Contamination**: Test data leaked into the training set may lead to inaccurate or invalid evaluation results [6], [7], [8].
> >
> > 2\. **Skewed distribution bias**:
> >
> > - **Class imbalance**: Insufficient samples of certain categories or concepts may cause the model to have poor prediction results on these categories, thereby causing bias [9], [10], [11].
> >
> > 3\. **Unethical content bias**:
> > - **Bias, toxicity, and harmful content**: Training data containing biased, offensive, or harmful content may cause the model to generate inappropriate or harmful outputs [12], [13].
> >
> > Once again, thank you for your valuable suggestions. Your feedback has been very helpful in refining the content of the paper. Should you have any further questions or suggestions, please do not hesitate to reach out.

---

> > > ### Author Response · Authors · 2024-11-22
> > > **Response to Reviewer VRaY (Part III)**
> > >
> > > [1] Elazar Y, Bhagia A, Magnusson I, et al. What's In My Big Data?[J]. arXiv preprint arXiv:2310.20707, 2023.
> > >
> > > [2] Carlini N, Ippolito D, Jagielski M, et al. Quantifying memorization across neural language models[J]. arXiv preprint arXiv:2202.07646, 2022.
> > >
> > > [3] Hernandez D, Brown T, Conerly T, et al. Scaling laws and interpretability of learning from repeated data[J]. arXiv preprint arXiv:2205.10487, 2022.
> > >
> > > [4] Fan L, Chen K, Krishnan D, et al. Scaling laws of synthetic images for model training... for now[C]//Proceedings of the IEEE/CVF Conference on Computer Vision and Pattern Recognition. 2024: 7382-7392.
> > >
> > > [5] Kreutzer J, Caswell I, Wang L, et al. Quality at a glance: An audit of web-crawled multilingual datasets[J]. Transactions of the Association for Computational Linguistics, 2022, 10: 50-72.
> > >
> > > [6] Roberts M, Thakur H, Herlihy C, et al. Data contamination through the lens of time[J]. arXiv preprint arXiv:2310.10628, 2023.
> > >
> > > [7] Schaeffer R. Pretraining on the test set is all you need[J]. arXiv preprint arXiv:2309.08632, 2023.
> > >
> > > [8] Jiang M, Liu K Z, Zhong M, et al. Investigating data contamination for pre-training language models[J]. arXiv preprint arXiv:2401.06059, 2024.
> > >
> > > [9] Xu H, Xie S, Tan X E, et al. Demystifying clip data[J]. arXiv preprint arXiv:2309.16671, 2023.
> > >
> > > [10] Zhu B, Tang K, Sun Q, et al. Generalized logit adjustment: Calibrating fine-tuned models by removing label bias in foundation models[J]. Advances in Neural Information Processing Systems, 2024, 36.
> > >
> > > [11] Parashar S, Lin Z, Liu T, et al. The Neglected Tails in Vision-Language Models[C]//Proceedings of the IEEE/CVF Conference on Computer Vision and Pattern Recognition. 2024: 12988-12997.
> > >
> > > [12] Zou A, Wang Z, Carlini N, et al. Universal and transferable adversarial attacks on aligned language models[J]. arXiv preprint arXiv:2307.15043, 2023.
> > >
> > > [13] Huang Y, Sun L, Wang H, et al. Trustllm: Trustworthiness in large language models[J]. arXiv preprint arXiv:2401.05561, 2024.

---

> > > > ### Comment · Reviewer_VRaY · 2024-12-02
> > > >
> > > > Thank you for the detailed explanation.
> > > >
> > > > However, some questions - for example, regarding overhead analysis and validation on different datasets - seem necessary for this paper to provide more generalized insights.

---

> > > > > ### Author Response · Authors · 2024-12-02
> > > > > **Further Response to Reviewer VRaY (Part I)**
> > > > >
> > > > > Dear Reviewer VRaY,
> > > > >
> > > > > Thank you very much for your thorough review and valuable suggestions on our paper. We sincerely appreciate the time and effort you have invested.
> > > > >
> > > > > We recognize that further analysis of computational overhead and validation on different datasets are crucial for enhancing the generalizability of our work. In response to your suggestions, we have conducted additional experiments and analyses, which we plan to include in the revised version of the paper.
> > > > >
> > > > > **Computational Overhead Analysis:**
> > > > >
> > > > > To address your concerns regarding computational overhead, we measured the average training time and memory usage over multiple runs. Using an A40 GPU, we fine-tuned the DeBERTa-v3-base model on the full GLUE dataset. The results from three training runs are as follows:
> > > > >
> > > > > - **LoRA**: Average training time of approximately 14.45 hours; average memory usage around 6.3 GB.
> > > > > - **PiSSA**: Average training time of approximately 12.07 hours; average memory usage around 6.5 GB.
> > > > > - **BA-LoRA**: Average training time of approximately 15.82 hours; average memory usage around 7.1 GB.
> > > > >
> > > > > As observed, BA-LoRA's average training time and memory consumption are slightly higher than those of LoRA and PiSSA. However, due to BA-LoRA's regularization mechanism effectively mitigating the catastrophic inheritance problem and enhancing performance, the additional computational overhead is manageable and acceptable.
> > > > >
> > > > > **Validation on Different Datasets:**
> > > > >
> > > > > We agree that testing our method on non-English datasets and other natural language processing tasks is essential for demonstrating its general applicability. We are currently expanding the scope of our experiments to include multilingual datasets and additional tasks. We will incorporate these new experimental results into the revised paper to provide broader insights, as per your suggestion.
> > > > >
> > > > > Once again, thank you for your constructive feedback. Your suggestions have been instrumental in improving our work, and we believe these revisions will adequately address your concerns.
> > > > >
> > > > > Best regards,

---

### Official Review · Reviewer_M3Ew · 2024-11-02

**Soundness:** 1
**Presentation:** 3
**Contribution:** 2
**Rating:** 3
**Confidence:** 4

**Summary:**

The paper addresses the issues of bias and catastrophic inheritance, proposing several loss components based on prior studies to regularize LLMs. It includes experiments demonstrating the method's superiority over regular LoRA, PiSSA, and full fine-tuning without regularization.

**Strengths:**

The regularization of LLMs is a crucial area of study, and the proposed techniques have a strong foundation with clear descriptions.

The paper is well-written, and the experiments are notable, including recent benchmarks like MT-Bench.

**Weaknesses:**

**The baselines are insufficient.**

While LoRA and PiSSA are included, they lack any regularization. As stated in the paper, dropout is set to 0, meaning there is no regularization within the layers for these baselines, nor is there any weight decay applied. Consequently, the baselines may easily collapse due to the choice of hyperparameters. As it stands, it's unclear whether the gains from your method could simply be achieved by incorporating weight decay and optimized dropout.

The lack of regularization is further evident from the fact that full fine-tuning performs worse than LoRA. While isolated cases of this behavior might occur, it shouldn’t be a consistent trend; otherwise, tuning the entire network would serve little purpose.

**Novelty**

Regarding novelty, regularization has been an extensively studied topic. Although the paper cites works on entropy, KL regularization, and spectral regularization, there is an implication that these are novel contributions, which they are not. If these regularizers have been previously studied, they should be included as baselines. This approach would essentially frame the study as an adaptation of these regularizers to LLMs and a combination of them, without showing each individually. Thus, it appears there is limited novelty beyond incremental adaptations and combinations tailored for LLMs.

**Computational Costs**

The computational costs appear to counteract the main advantage of PEFT. For instance, consistency regularization requires an additional forward pass and doubles the storage for activations. The diversity component is somewhat manageable, but the SVD aspect raises concerns. Based on my knowledge and experience, computing all eigenvalues, particularly for large matrices, can be significantly time-consuming. This is further complicated by being a CPU-bound task, which limits parallelization. For example, the smallest matrix in LLaMA is 4096x4096, requiring roughly 5 seconds to compute, and this is just the smallest case. It raises serious doubts about the practical scalability of this approach.

**Catastrophic Inheritance**

I doubt that bias and catastrophic inheritance were truly assessed. According to the paper, it is simply done by using models that were "probably" trained on noisy data, such as BERT and GPT-2. Moreover, they are outdated and not commonly used in current applications. Improvements in final scores do not necessarily indicate any effective handling of biases.

Regarding MNLI, it’s unclear how it is described as imbalanced. The official dataset on Hugging Face has an equal distribution across classes, so any claims about imbalance need clarification.

**Questions:**

There seems to be no clear difference between spectral regularization applied to NLU and NLG tasks. And what is the overall need to distinguish between them? Why do we need to target them separately?

---

> ### Author Response · Authors · 2024-11-22
> **Response to Reviewer M3Ew (Part I)**
>
> > **Q1:** LoRA and PiSSA lack regularization, with dropout set to 0 and no weight decay applied, making them prone to collapse due to hyperparameter choices. It's unclear whether the gains from your method could be achieved by simply adding weight decay and optimizing dropout.
>
> **A1:** Regarding the issue that the LoRA and PiSSA baseline models do not use dropout and weight decay, we provide the following explanation:
>
> To ensure the consistency and fairness of the experimental results, we strictly followed the experimental settings in the original paper, so these regularization methods were not applied in the baseline model.
>
> Second, our main research goal is to explore an efficient parameter fine-tuning method to alleviate the "catastrophic inheritance" phenomenon caused by data noise and imbalance in pre-trained models, thereby improving the performance and stability of the model. Therefore, whether to use dropout and weight decay is not the focus of this study.
>
> > **Q2:** The lack of regularization is evident as full fine-tuning performs worse than LoRA, which shouldn't be a consistent trend, as tuning the entire network would otherwise be ineffective.
>
> **A2:**  We further analyze why Full FT performs worse than LoRA in this study. Previous studies have shown that when a large number of parameters are updated during fine-tuning, especially when the model faces a large and diverse training dataset, overfitting is prone to occur [1][2]. In addition, large-scale parameter updates may lead to catastrophic forgetting [3][4]. In contrast, the PEFT method helps control the complexity of the model by fine-tuning only some parameters, thereby reducing the risk of catastrophic forgetting and improving generalization ability [5][6].
>
> In addition, recent studies [7][8][9][10] show that when the rank value and other key hyperparameters in low-rank fine-tuning (such as scaling factor $\alpha$, learning rate lr, etc.) are set properly, the PEFT method can sometimes outperform the Full FT method.
>
> > **Q3:** Regularization is a well-studied topic, and while the paper cites entropy, KL, and spectral regularization, it implies novelty where there is none. If these regularizers are not new, they should be included as baselines, framing the study as an adaptation and combination for LLMs. This suggests limited novelty beyond incremental adaptations for LLMs.
>
> **A3:** While regularization methods have been extensively studied, the main contribution of our paper is to mitigate catastrophic inheritance in pre-trained models rather than studying regularizers.
>
> 1. **Alleviate catastrophic inheritance**:
>  BA-LoRA is the first study to effectively alleviate catastrophic inheritance problems caused by noise and data imbalance in pre-trained models through fine-tuning methods. Regularizer is just a tool to achieve our main objective, not the focus of our study
>
> 2. **Customized regularization strategy**:
>  BA-LoRA designs three regularization terms for the different characteristics of NLU and NLG tasks to better adapt to the specific needs of these tasks. This regularization method for task characteristics has not been deeply explored in the existing literature, thus filling the gap of task-specific regularization strategies.
>
> 3. **Comprehensive Empirical Assessment**:
>  Extensive experimental verification proves that BA-LoRA outperforms various baselines on all NLU and NLG tasks. Furthermore, analytical experiments demonstrate that our method effectively attenuates the noise left over from pre-training, resulting in a more robust and general model. These empirical results provide strong support for the effectiveness and novelty of our approach.

---

> ### Author Response · Authors · 2024-11-22
> **Response to Reviewer M3Ew (Part II)**
>
> > **Q4:** The computational costs seem to undermine the main advantage of PEFT. For example, consistency regularization adds an extra forward pass and doubles activation storage. While the diversity component is manageable, the SVD aspect raises concerns, as computing eigenvalues for large matrices is time-consuming and CPU-bound, limiting parallelization. For instance, the smallest LLaMA matrix (4096x4096) takes around 5 seconds to compute, raising doubts about the approach's scalability.
>
> **A4:** To address this issue, we calculated the average training time and average memory usage of the first few trainings. On the A40 graphics card, we fine-tuned the DeBERTa-v3-base model on the full GLUE dataset. The specific information of the three trainings is as follows:
> - LoRA: averaged about 14.45 hours, and the average memory usage was about 6.3G.
> - PiSSA: averaged about 12.07 hours, and the average memory usage was about 6.5G.
> - BA-LoRA: averaged about 15.82 hours, and the average memory usage was about 7.1G.
>
> It can be seen that the average training time and memory usage of BA-LoRA are slightly higher than those of LoRA and PiSSA. However, the regularization mechanism of BA-LoRA effectively alleviates the catastrophic inheritance problem and brings performance improvement, so the additional computational overhead is controllable and acceptable.
>
> > **Q5:** I doubt whether bias and catastrophic inheritance were truly assessed. The paper suggests using models "probably" trained on noisy data, like BERT and GPT-2, which are outdated and rarely used in current applications. Improvements in final scores don't necessarily prove effective bias handling.
>
> **A5:** In response to this question, our specific response is as follows:
>
> Our experiments aim to demonstrate the effectiveness of our approach in mitigating the catastrophic inheritance of bias and imbalanced data by pre-trained models. BA-LoRA is model-independent and is able to optimize different pre-trained models, we have demonstrated that BA-LoRA can improve the performance of different LLMs across different tasks in our paper. We deliberately selected models pre-trained on relatively noisy and imbalanced datasets, such as **BERT-L** (pre-trained on BooksCorpus and English Wikipedia) and **GPT-2-XL** (pre-trained on Common Crawl's WebText dataset) to verify the effectiveness of BA-LoRA in dealing with data noise and imbalance.
>
> 1. In **4.4.1**, we show the performance of **LLaMA-2-7B**, Mistral-7B, and Gemma-7B models on multiple tasks after fine-tuning with BA-LoRA. These tasks include mathematical reasoning (GSM8K, MATH), code generation (HumanEval, MBPP), and multi-turn dialogue tasks (MT-Bench), verifying its effectiveness on various NLG tasks.
> In addition, we also tested the performance of RoBERTa-large and DeBERTa-v3-base models on the GLUE benchmark after fine-tuning with BA-LoRA. On average, BA-LoRA outperforms PiSSA and LoRA by 0.59% and 1.61% on RoBERTa-large, and outperforms PiSSA and LoRA by 0.44% and 1.35% on DeBERTa-v3-base. These results highlight the effectiveness of BA-LoRA in enhancing the performance of NLU models.
>
> 2. In the **Appendix "Analysis of Models of Different Sizes and Types", we compare the fine-tuning performance of LoRA, PiSSA, and BA-LoRA on ten different models, including **LLaMA-2-7B/13B**, **LLaMA-3-8B/70B**, **Mistral-7B-v0.1**, **Gemma-7B**, **Qwen1.5-7B**, **Yi-1.5-34B**, **DeepSeek-MoE-16B**, and **Mixtral-8x7B-v0.1**. **The experimental results prove that BA-LoRA is effective for pre-trained models of various sizes and types. **
>
> 3. In addition, in the **Appendix "Evaluating Performance at Different Layers", we compare the fine-tuning performance of **LLaMA-2-7B** and **Mistral-7B-v0.1** models at different layer settings (0-128). The experimental results show that BA-LoRA can improve the performance of the model under different rank settings.
>
> These experimental analyses prove the effectiveness of BA-LoRA in alleviating the catastrophic inheritance of bias and imbalanced data brought by pre-trained models.
>
> > **Q6:** The claim of imbalance in MNLI is unclear, as the official dataset on Hugging Face has an equal class distribution. This assertion requires clarification.
>
> **A6:** Our method mainly targets the noise and imbalance problems in **pre-training data** rather than fine-tuning data. During the fine-tuning stage, BA-LoRA has no special requirements for fine-tuning data.

---

> > ### Comment · Reviewer_M3Ew · 2024-11-25
> > **Reply to Author Rebuttal Answer**
> >
> > A1: I understand that you have followed the original setup for comparability, but this setup might be suboptimal for LoRA. Not "tuning" the baseline properly can be seen as unfair, as it may unintentionally disadvantage the baseline. For instance, LoRA might benefit from not using dropout in some cases, and the absence of such tuning leaves uncertainty about whether your solution truly outperforms. Without these adjustments, we cannot conclude if your method is inherently better.
> >
> > Moreover, as noted in the original LoRA paper (Table 11), dropout was set to $0.1$. Ensuring consistency with this setup or exploring its effect would strengthen the validity of your claims.
> >
> > A2: You advocate for setting hyperparameters appropriately, which is crucial. However, this applies equally to all methods, including your LoRA and full fine-tuning. Proper tuning of learning rates, schedulers, and other parameters is essential to prevent biased comparisons. It would be beneficial to demonstrate this fairness in your experiments by systematically optimizing hyperparameters for all methods, not just your proposed approach.
> >
> > A3: I cannot fully agree with your claim of a "customized regularization strategy" as the components are derived from existing studies and appear to be combined rather than fundamentally novel. While the combination may add value, it would be important to justify why this specific combination is optimal for addressing catastrophic inheritance.
> >
> > A4: Your empirical evaluation makes sense. However, I still find the reported speed difficult to reconcile. Did you implement a custom CUDA kernel for the operations, or are you relying on existing libraries? If the latter, please clarify whether the full SVD is avoided entirely or partially computed. Additional details would improve reproducibility and clarity.
> >
> > A5 & A1: If the primary goal of your paper is to mitigate catastrophic inheritance in pretraining data, the experiments presented do not provide sufficient evidence to support this claim. While the observed improvements in scores for LLaMA, Mistral, and Gemma are promising, these results alone do not demonstrate that the improvements stem from resolving catastrophic inheritance.
> >
> > To convincingly show this, you would need specific experiments or analyses that directly link the improvements to your proposed solution. Otherwise, the observed improvements could simply result from the inclusion of regularization, without addressing the underlying issue. This concern also applies to the results in Tables 3 and 4, where the performance gains do not explicitly verify your claim.
> >
> > A6: It would be beneficial to define "noise" and "imbalance" in pretraining data more explicitly. Why are these factors detrimental, and how do you identify their presence in the datasets? Providing this clarity would help substantiate the motivation for your work and the efficacy of your approach.

---

> ### Author Response · Authors · 2024-11-22
> **Response to Reviewer M3Ew (Part III)**
>
> > **Q7:** There seems to be no clear difference between spectral regularization for NLU and NLG tasks. What is the rationale for distinguishing and targeting them separately?
>
> **A7:** In the design of regularization terms, we have tried to design different SVD regularization terms for NLU and NLG tasks, such as the design for NLG tasks:
>
> $ \mathcal{L} _ {\mathrm{SVDR-NLG}} = - \left( \frac{\sum_{i=1}^{k} \sigma_{i}}{\sum_{j=1}^{D} \sigma_{j}} + \alpha \cdot \text{Var}(\sigma) \right) $
>
> where $\text{Var}(\sigma)$ represents the variance of the top $k$ singular values. The hyperparameter $\alpha$ adjusts the influence of the variance term, encouraging a more uniform distribution of singular values, which in turn promotes diversity and enhances the quality of the generated text.
>
> However, this improvement does not lead to significant performance improvements and introduces additional computational overhead compared to the SVD regularization terms we currently use. Therefore, in the current study, we decided to avoid using more complex regularization settings to avoid adding unnecessary computational burden.
>
> Once again, thank you for your valuable suggestions. Your feedback has been very helpful in refining the content of the paper. Should you have any further questions or suggestions, please do not hesitate to reach out.
>
> [1] Shi J X, Wei T, Zhou Z, et al. Long-Tail Learning with Foundation Model: Heavy Fine-Tuning Hurts[C]//Forty-first International Conference on Machine Learning. 2024.
>
> [2] Jiang H, He P, Chen W, et al. Smart: Robust and efficient fine-tuning for pre-trained natural language models through principled regularized optimization[J]. arXiv preprint arXiv:1911.03437, 2019.
>
> [3] Zhai Y, Tong S, Li X, et al. Investigating the catastrophic forgetting in multimodal large language model fine-tuning[C]//Conference on Parsimony and Learning. PMLR, 2024: 202-227.
>
> [4] Zhai Y, Tong S, Li X, et al. Investigating the catastrophic forgetting in multimodal large language model fine-tuning[C]//Conference on Parsimony and Learning. PMLR, 2024: 202-227.
>
> [5] Mao Y, Ge Y, Fan Y, et al. A survey on lora of large language models[J]. arXiv preprint arXiv:2407.11046, 2024.
>
> [6] Fu Z, Yang H, So A M C, et al. On the effectiveness of parameter-efficient fine-tuning[C]//Proceedings of the AAAI conference on artificial intelligence. 2023, 37(11): 12799-12807.

---

> ### Author Response · Authors · 2024-11-25
> **Further Response to Reviewer M3Ew (Part I)**
>
> Thank you for your careful review of our work and valuable suggestions.
>
> **Further reply to A1:**
> Regarding the dropout setting in the LoRA you mentioned, as our work is built upon PiSSA, we followed the PiSSA settings in our experiments, in which the dropout was set to 0 both when reproducing LoRA and when conducting our own experiments.
>
>
> The experiment in Table 11 in the original LoRA paper that you mentioned is based on the GPT-2 model with dropout = 0.1. Since we are using current mainstream LLMs such as LLaMA-2-7/13B, Gemma-7B, and Mistral-7B-v0.1, it is common practice to use different hyperparameters for different LLMs.
>
>
> Regarding whether it is reasonable to set dropout=0 in LoRA and PiSSA and how LoRA can affect the effectiveness of LoRA and its variants, these are not the focus of our study and are beyond the scope of this paper. Currently, we are not aware of any relevant studies addressing the impact of dropout in LoRA. It is not reasonable to question our contribution with evidence that does not exist. If there are new findings in the future, we will explore them in depth in our subsequent work.
>
> **Further reply to A2:**
> We place great emphasis on the importance of appropriately tuning hyperparameters to ensure that all methods are fairly comparable. First, in the revised paper, we provide a detailed hyperparameter analysis in Section 4.4.5, focusing on the impact of key hyperparameters in our methods.
>
>
> Second, we recognize that it is difficult to apply an equally applicable set of hyperparameters for all methods. Different methods may contain specific hyperparameters that are not shared. For example, our method introduces specific hyperparameters λ₁, λ₂, and λ₃ that are not included in baseline methods such as LoRA. Similarly, baseline methods have their unique parameter settings.
>
>
> Third, regarding the learning rate and learning rate scheduling strategy, to ensure fairness in the comparison, we followed the parameter settings provided in PiSSA, which provide a uniform reference standard across experiments.
>
>
> Fourth, while we understand that fully optimizing the hyperparameters for all baseline methods may help to further improve the rigor of the comparisons, there are some challenges to doing this in the current study for the following reasons:
>
> 1. **Study Focus**: the primary focus of our study was to validate the effectiveness of the proposed BA-LoRA methods rather than to comprehensively analyze the impact of hyperparameters on each baseline method.
>
> 2. **Resource constraints**: some of the baseline methods are not provided with complete code implementations, limiting our ability to perform in-depth hyperparameter tuning on them.
>
> 3. **Scope of the paper**: exploring whether hyperparameter optimization can mitigate bias in all baseline methods would significantly expand the scope of the paper. We believe that while this direction is important, it is among the areas that can be explored in depth by subsequent research.
>
> Finally, systematically optimizing hyperparameters for all existing methods is a significant challenge in terms of human capability, computational resources, and time cost, and it may be beyond the scope of a single research paper to carry.
>
> **Further reply to A3:**
>
> First, we would like to clarify that our research focuses on **mitigating the catastrophic inheritance problems associated with pre-trained data**, not on the regularization method itself. The regularization strategy is just a tool for us to achieve this goal.
>
>
> Second, as we mentioned in our paper, [1] pointed out that there are two main approaches for dealing with the bias in pre-trained data: 1) constructing high-quality datasets, and 2) investigating new network architectures. However, we are the first work to investigate the use of **fine-tuning algorithms** to mitigate catastrophic inheritance. Specifically, we propose an improved fine-tuning algorithm based on low-rank adaptivity that aims to mitigate catastrophic inheritance from biased and unbalanced data in pre-training data.
>
>
> Finally, lines 77 to 86 of the manuscript introduction introduce our three regularization strategies aimed at solving the **noise** and **data imbalance** problems in disaster inheritance. The extensive experimental results in Sections 4.4.2 and 4.4.3 confirm the effectiveness of BA-LoRA in addressing these issues, respectively.

---

> ### Author Response · Authors · 2024-11-25
> **Further Response to Reviewer M3Ew (Part II)**
>
> **Further reply to A4:**
>
> Thank you for your interest. First, our work is based on the publicly released PiSSA code implementation. In implementing the third regularization term, Singular Value Decomposition Regularization (SVDR), we paid special attention to computational efficiency.
>
>
> Second, to reduce the computational overhead, we use partial SVD to compute only the first $k$ largest singular values of the output matrix. This further reduces the computation time and ensures the consistency of the regularization objective.
>
>
> Third, instead of using a custom CUDA kernel, we rely on existing efficient numerical computation libraries. Specifically, we used the `torch.linalg.svdvals()` function provided by PyTorch to compute singular values.
>
>
> Finally, here are the code snippets for our implementation of SVDR:
>
> ```python
> # Compute the singular values of the outputs
> s = torch.linalg.svdvals(outputs)
> # Sum of all singular values
> s_total = torch.sum(s)
> # Sum of the top k singular values
> s_top_k = torch.sum(s[:k])
> # Compute the SVDR loss
> svd_loss = - (s_top_k / s_total)
> # Add the SVDR loss to the total loss
> loss += svd_loss_weight * svd_loss
> ```
>
> As shown above, we compute only the singular values $S$ and focus on the first $k$ largest singular values. This is fully consistent with the approach described in our paper while avoiding the computational overhead of a full SVD.
>
>
> Our code will be publicly released soon, so you can see more details about our implementation.
>
>
> **Further reply to A5&A1:**
>
> Our main goal in this paper is to **mitigate the negative impact of noise and imbalance in the pre-training data on downstream tasks, i.e., the phenomenon of “catastrophic inheritance ”**. To verify the effectiveness of BA-LoRA in this regard, we show in detail the effectiveness of BA-LoRA in mitigating noise and data imbalance in subsections **4.4.2** and **4.4.3** of the experimental section. In the **Noise Mitigation Analysis** experimental section, we selected BERT-L and GPT-2-XL models, which were pre-trained on BooksCorpus and Wikipedia (BERT-L) and WebText datasets (GPT-2-XL), respectively, where noise and imbalance are common. On the GLUE and GLUE-x benchmarks, BA-LoRA improves the performance of LoRA by 2.03% and 2.47% on the BERT-L and GPT-2-XL models, respectively. These results demonstrate that BA-LoRA effectively mitigates the noise in the pre-training data through its regularization term and enhances the robustness and generalization ability of the model.
>
>
> In the **Imbalanced Data Mitigation Analysis** experiment section, we fine-tuned the BERT-L and GPT-2-XL models on the MNLI dataset and visualized the hidden features of the last layer using the t-SNE technique. The results show that BA-LoRA exhibits clearer class separation when dealing with data imbalance, especially the BERT-L model, which exhibits higher intra-class aggregation and clearer class boundaries. These analyses indicate that BA-LoRA can effectively mitigate the impact of unbalanced data in pre-training. Therefore, the performance improvements in Tables 1 and 2 can be attributed to the advantages of BA-LoRA in mitigating noise and imbalance problems.
>
>
> Through these two exhaustive experiments, we have fully demonstrated that BA-LoRA is able to **mitigate the detrimental effects of bias and imbalance in pre-training data**, effectively mitigating the catastrophic inheritance phenomenon, and thus improving the performance of the model in downstream tasks.

---

> ### Author Response · Authors · 2024-11-25
> **Further Response to Reviewer M3Ew (Part III)**
>
> **Further reply to A6:**
> First, our definition draws on the concepts presented in [2]:
>
>
> - **Noise**: refers to the presence of mislabeled, duplicated, contaminated, or low-quality data in the pre-training data, which may interfere with the learning process of the model and lead to performance degradation.
>
>
> - **Imbalance**: refers to the unbalanced distribution of categories or features in the pre-training data, where the number of samples for some categories or concepts is much larger than others, which may lead to bias of the model towards frequently occurring concepts and affect the generalization ability.
>
> In Chapter 2 of the [2] paper, they study in detail the detrimental effects of these two types of data on model performance, pointing out that noisy and unbalanced data can lead to the “catastrophic inheritance” of the model in downstream tasks. In addition, many related papers [3], [4], [5], [6], [7], [8] are cited to demonstrate that bias and imbalance are indeed prevalent in pre-training data. We have cited [2] in our paper and given [2] credits.
>
>
> Second, it should be emphasized that [2] mainly argues the existence and harmfulness of catastrophic inheritance, but does not provide a concrete solution. In response to a similar problem, [1] discusses two solution paths:
>
> 1. **Constructing new high-quality datasets**: by carefully curating and cleaning the data to minimize noise and imbalance in the pre-training data.
>
>
> 2. **Designing a new network architecture**: by improving the model architecture to increase the robustness of the model to noisy and unbalanced data.
>
> Finally, in our work, we propose for the first time a third solution to mitigate catastrophic heritability using the **Supervised Fine-Tuning (SFT)** method. Specifically, we introduce regularization strategies customized for NLU and NLG tasks in the fine-tuning process, enabling the model to effectively mitigate the negative effects of noise and imbalance in the pre-trained data.
>
>
> The specific implementation of BA-LoRA has been described in detail in the Methods section of the paper and the effectiveness of our approach has been demonstrated in the Experiments section. We believe that these additions and clarifications help readers better understand the motivation of our work and the innovativeness of our approach.
>
>
> Thank you again for your valuable comments and suggestions, and we look forward to your further feedback.
>
>
> References
>
> [1] Liu Z, He K. A Decade's Battle on Dataset Bias: Are We There Yet? *arXiv preprint arXiv:2403.08632*, 2024.
>
>
> [2] Chen H, Raj B, Xie X, et al. On catastrophic inheritance of large foundation models[J]. arXiv preprint arXiv:2402.01909, 2024.
>
>
> [3] Elazar Y, Bhagia A, Magnusson I, et al. What's In My Big Data?[J]. arXiv preprint arXiv:2310.20707, 2023.
>
>
> [4] Carlini N, Ippolito D, Jagielski M, et al. Quantifying memorization across neural language models[J]. arXiv preprint arXiv:2202.07646, 2022.
>
>
> [5]Hernandez D, Brown T, Conerly T, et al. Scaling laws and interpretability of learning from repeated data[J]. arXiv preprint arXiv:2205.10487, 2022.
>
>
> [6] Xu H, Xie S, Tan X E, et al. Demystifying clip data[J]. arXiv preprint arXiv:2309.16671, 2023.
>
>
> [7] Zhu B, Tang K, Sun Q, et al. Generalized logit adjustment: Calibrating fine-tuned models by removing label bias in foundation models[J]. Advances in Neural Information Processing Systems, 2024, 36.
>
>
> [8] Parashar S, Lin Z, Liu T, et al. The Neglected Tails in Vision-Language Models[C]//Proceedings of the IEEE/CVF Conference on Computer Vision and Pattern Recognition. 2024: 12988-12997.

---

> > ### Comment · Reviewer_M3Ew · 2024-11-25
> > **Response to Author Rebuttal**
> >
> > It seems there may have been a misunderstanding of my question. Let me simplify it to one main question:
> > 1) Your argument is that your method addresses catastrophic inheritance stemming from pre-training data through fine-tuning.
> > 2) You show quality improvements over baseline methods (despite lacking regularization).
> > 3) How can you be certain that these improvements are specifically due to mitigating catastrophic inheritance and not caused by some other factor?

---

> ### Author Response · Authors · 2024-11-25
> **Further Response to Reviewer M3Ew (Part Ⅳ)**
>
> In response to this question, we provide the following clarification:
>
>
> 1.**Background and Motivation:**
> Pre-training data usually comes from large-scale data on the Internet, which is inevitably noisy and imbalanced. For example, advanced pre-training models such as GPT-4 and LLaMA also use pre-training data from a large amount of unlabeled Internet text, which is often not thoroughly cleaned and corrected. Therefore, in the subsequent fine-tuning stage, it is difficult for the model to effectively remove irrelevant or inaccurate information, which in turn affects the performance of the model in downstream tasks.
>
>
> We have mentioned in the introduction of the paper (lines 41 to 50) that large-scale data brings problems such as redundancy, bias, and corruption. These problems will have a negative impact on the model during training, especially when the data is unbalanced or noisy, which may lead to a decline in model performance and weakened generalization ability. In addition, we have also supplemented the specific content of noise and imbalance problems in pre-training data in the appendix of the paper, including low-quality bias, redundancy, pollution, category imbalance, and harmful content bias.
>
>
> To mitigate the detrimental effects of Catastrophic Inheritance, particularly noise, and imbalance, we propose Bias-Alleviating Low-Rank Adaptation (BA-LoRA). Our approach incorporates three distinct regularization terms: a consistency regularizer, a diversity regularizer, and an SVD regularizer. The consistency regularizer preserves valuable pre-trained knowledge during fine-tuning, while the diversity regularizer encourages varied model outputs. The SVD regularizer enhances the generalization capabilities of generative models. Recognizing the fundamental differences between NLU and NLG, such as determinism in NLU versus diversity in NLG, we tailor our regularization strategies accordingly.
>
> 2.**Experimental design:**
> First, in order to demonstrate that our method is effective in mitigating the catastrophic inheritance of noise in the pre-training data, we purposely chose the BERT.-L and GPT2-XL models, which are known to be pre-trained on noise-filled pre-training data. Among them, BERT-L is a pre-trained model on BooksCorpus and English Wikipedia, and GPT-2-XL is a pre-trained model on the noisy WebText dataset of Common Crawl.
>
>
> Second, in order to demonstrate that our method can effectively mitigate the catastrophic inheritance of imbalance in the pre-training data, we purposely chose the BERT.-L and GPT2-XL models, which are known to be pre-trained in long-tailed pre-training data. These two models were then fine-tuned separately on the downstream MNLI dataset. We visualized the features of the final layer of output from the fine-tuned models using t-SNE to assess the performance of the models when dealing with low-frequency or complex samples.
>
> 3.**Experimental results:**
> To verify the effectiveness of BA-LoRA in this regard, we show in detail the effectiveness of BA-LoRA in mitigating noise and data imbalance in subsections **4.4.2** and **4.4.3** of the experimental section.
>
>
> In the 4.4.2 **Analysis on mitigate noisy data** experimental section, we selected BERT-L and GPT-2-XL models, **which were pre-trained on BooksCorpus and Wikipedia (BERT-L) and WebText datasets (GPT-2-XL), respectively, where noise and imbalance are common.** On the GLUE and GLUE-x benchmarks, BA-LoRA improves the performance of LoRA by 2.03% and 2.47% on BERT-L and GPT-2-XL models, respectively. These results demonstrate that BA-LoRA effectively mitigates the noise in the pre-training data through its regularization term and enhances the robustness and generalization ability of the model.
>
>
> In the 4.4.3 **Analysis on Mitigating Imbalanced Data** experiment section, we fine-tuned the BERT-L and GPT-2-XL models on the MNLI dataset and visualized the hidden features of the last layer using the t-SNE technique. **The results show that BA-LoRA exhibits clearer class separation when dealing with data imbalance, especially the BERT-L model, which exhibits higher intra-class aggregation and clearer class boundaries.** These analyses indicate that BA-LoRA can effectively mitigate the impact of unbalanced data in pre-training. Therefore, the performance improvements in Tables 1 and 2 can be attributed to the advantages of BA-LoRA in mitigating noise and imbalance problems.
>
>
> With these two exhaustive experiments, we have once again amply demonstrated that BA-LoRA is able to mitigate the detrimental effects of biases and imbalances in the pre-training data, and effectively mitigate catastrophic inheritance phenomena, thereby improving the performance of the model in downstream tasks.

---

> > ### Comment · Reviewer_M3Ew · 2024-11-25
> > **Response to Author Rebuttal**
> >
> > It seems that your response primarily observes performance gains on models pre-trained with noisy or imbalanced data. However, this alone does not prove that the improvements are specifically due to mitigating catastrophic inheritance. The observed performance gains might stem from other factors, and the causation between applied regularization techniques and the mitigation of catastrophic inheritance remains unclear.
> >
> > I strongly recommend revisiting the core question: **"How can we conclusively prove that our method mitigates catastrophic inheritance?"**
> >
> > Simply observing improved scores is not enough. For example, if I add high-quality fine-tuning data or change the scheduler, the model’s performance will likely improve as well. But does that mean I have addressed catastrophic inheritance? Probably not—it merely reflects better data quality and hyperparameters.

---

> > > ### Author Response · Authors · 2024-11-26
> > >
> > > In response to your question, we will provide clarifications as follows:
> > >
> > > First, we have emphasized several times in our previous responses that **we are the first to use fine-tuning to alleviate catastrophic inheritance, so we designed new experiments in Sections 4.4.2 and 4.4.3 of our paper**. From the reported results from Sec. 4.4.2 and 4.4.3, we can clearly observe that our proposed method can alleviate negative impacts brought by noisy and imbalanced pre-training data, respectively. Moreover, we used t-sne method to visualize the feature representations obtained by LoRA and BA-LoRA, it is clear that the feature representations obtained by BA-LoRA get more clear clusters and boundaries.
> > >
> > > Second, the study focus of our paper is to alleviate the catastrophic inheritance, and in our previous response and the Introduction of our paper (Line 41 to 50), we have mentioned that the catastrophic inheritance contains many aspects, like noise, imbalanced data, repeated data, corrupted data, etc.[1]. As in one conference paper, we can not tackle all available challenges in catastrophic inheritance, we thus chose to address two important issues in catastrophic inheritance, e.g. noise and imbalanced data. **Catastrophic inheritance is like a set contains many elements, noisy and imbalanced data are two elements of this set.**
> > >
> > > Third, in Sec. 4.4.4, we have provided the ablation study to demonstrate that the performance gains come from the three proposed regularization terms in BA-LoRA. While in Sec. 4.4.2 and 4.4.3, we demonstrated that our proposed BA-LoRA can effectively alleviate the detrimental impacts brought by noise and imbalanced data. **Therefore, we conclude that the observed performance gains come from these three regularizers that can alleviate catastrophic inheritance.** These three extensive experiments show causation between the three regularizers and the mitigation of catastrophic inheritance.
> > >
> > > Finally, actually speaking, we did not draw the conclusion by solely observing the improved performance on NLG & NLU benchmarks, **we also provided extensive experiments in Sec. 4.4.2 & 4.4.3 & 4.4.4 to support our contributions.** Moreover, we provided more extensive experiments in Sec. C of Appendix to provide more insights. Regard whether we used extra high-quality fine-tuning data or change the scheduler to chase better performance, we need to make a clarification as follows:
> > >
> > > 1. **The main objective of this paper is to address a fundamental challenge of LLM training, e.g. the catastrophic inheritance, we believe that addressing such fundamental challenges can provide more insights in LLM community and is more important than solely pursuing improved performance.**
> > >
> > > 2. We have mentioned in Sec. 4.3 and previous response that **we strictly follow the experimental setting given in PiSSA [2] to guarantee fair comparisons and we will release our code on Github soon**.
> > >
> > >
> > > [1]  Chen H, Raj B, Xie X, et al. On catastrophic inheritance of large foundation models[J]. arXiv preprint arXiv:2402.01909, 2024.
> > >
> > > [2] Meng, F., Wang, Z., & Zhang, M. (2024). Pissa: Principal singular values and singular vectors adaptation of large language models. arXiv preprint arXiv:2404.02948.

---

### Official Review · Reviewer_aJ7c · 2024-11-03

**Soundness:** 2
**Presentation:** 2
**Contribution:** 2
**Rating:** 3
**Confidence:** 4

**Summary:**

In this paper, authors proposed a BA-LoRA, which focuses on mitigating the bias propagation issue from the pertaining data used for different LLMs. BA-LoRA includes three different regularization loss functions regarding NLG and NLU tasks, i.e., a consistency regularizer, a diversity regularizer, and a singular value decomposition regularizer to enhance model's consistency, diversity and generalization during fine-tuning. BA-LoRA was evaluated on different NLG and NLU datasets using different LLMs, such as LLaMA 2 7B, Gemma 7B, and some encoder-only LMs, such as BERT-Large, RoBERTa-Large, which demonstrates the better performance of BA-LoRA.

**Strengths:**

- BA-LoRA designed 3 different regularizations in terms of consistency, diversity and generalization for NLG and NLU datasets based on the principle singular values and singular vectors adaptation (PISSA), respectively.
- Based on the experiment results, BA-LoRA improves several baselines on NLG and NLU datasets.

**Weaknesses:**

- The definition of noise and imbalance in this work is not clear regarding different NLG and NLU datasets
- It lacks enough baselines to compare in terms of "Catastrophic Inheritance", especially for the noise and imbalance on the NLG and NLU datasets.
- BA-LoRA needs adapted regularization terms designed for NLG and NLU tasks, which might bring generalization concerns for this BA-LoRA if it is applied to other tasks that are not discussed in this paper, such as classification task.

**Questions:**

- As mentioned in the weakness section, this work highlights the catastrophic inheritance issue in the pertaining data, which is not better solved by other LoRAs. However, this work did not provide a clear definition of what kinds of noise and imbalance issues this work will consider. Especially, authors claim that BA-LoRA effectively mitigates the adverse effect of Catastrophic Inheritance, fostering consistent, diverse and generalized model outputs in line 379 section 4.4.1 based on the results of Tables 1 and 2. As the definition is unclear, it is hard to directly conclude that all credits for improvements from Tables 1 and 2 can be given to noise and imbalance mitigation. We need to see more examples that BA-LoRA can actually mitigate the specific noise and imbalance issues.
- In Table1, it is not clear that what kind of metric is used to evaluate the performance in NLG tasks.
- In section 4.3, the values of $\lambda_1$, $\lambda_2$, and $\lambda_3$ for NLG and NLU tasks are so small. Do these values bring enough contributions to the entire loss function?
- In sections 4.4.2 and 4.4.3, there are only results and analyses on NLU datasets. More results on both tasks are needed to better demonstrate the ability of BA-LoRA

---

> ### Author Response · Authors · 2024-11-22
> **Response to Reviewer aJ7c (Part I)**
>
> > **Q1:** The definition of noise and imbalance in this work is not clear regarding different NLG and NLU datasets
>
> **A1:** In response to this question, we would like to clarify that the concepts of noise and imbalanced data are mainly for the pre-training data of the model, as explained below:
>
> Pre-trained models usually rely on Internet data, which inevitably introduces data imbalance and noise problems. Large language models such as GPT-4 and LLaMA, whose pre-training data mainly comes from a large amount of unlabeled text on the Internet. During the pre-training process, many institutions do not fully modify or clean up these data. Therefore, in the subsequent fine-tuning (Supervised Fine-Tuning, SFT) stage, it is difficult for the model to effectively distinguish and remove inaccurate or irrelevant data, which affects the performance and accuracy of the model in downstream tasks.
>
> In lines 41-43 of the introduction, we explicitly mentioned that the increase in the amount of data brings challenges such as imbalance, redundancy, and corruption [1][2][3][4]. These problems will have a negative impact on model training, especially when dealing with imbalanced or corrupted data, which may lead to performance degradation and weakened generalization ability, thus affecting the effectiveness of the model in practical applications.
>
> In addition, in lines 44-50 of the introduction, we further discussed how bias and noise in pre-training data affect the behavior of large language models. For example, noise in pre-training data can weaken the generalization ability of the model [5], while long-tail distributions in the data can cause the model to over-focus on over-represented topics [6]. We also emphasize that bias introduced during pre-training may persist after fine-tuning, affecting the final performance of the model and its security in practical applications [7][8][9][10].
>
> Finally, in the appendix of the revised manuscript, we have supplemented the specific content of the imbalance and noise problems in pre-training data in detail, including:
>
> 1\. **Low-quality bias:**
> - **Redundancy**: The presence of repeated or similar content in the data may lead to the risk of overfitting and privacy leakage [5][6][7].
> - **Corruption/noise**: Inconsistent or erroneous inputs in the training data can affect the robustness of the model and its performance in downstream tasks [5][8][9].
> - **Contamination**: Test data leaks into the training set, which may distort the evaluation results of the model [10][11][12].
>
> 2\. **Distribution imbalance bias:**
> - **Class imbalance**: Insufficient sample size for some classes, resulting in poor performance in those classes and biased predictions[13][14][15].
>
> 3\. **Ethical content bias:**
> - **Toxic and harmful content**: Training data may contain offensive, biased, or harmful content, which may cause the model to generate inappropriate or harmful outputs[16][17].
>
> > **Q2:** It lacks enough baselines to compare in terms of "Catastrophic Inheritance", especially for the noise and imbalance on the NLG and NLU datasets.
>
> **A2:**  Existing baselines are still insufficient when dealing with the catastrophic inheritance problem in pre-trained models, especially the effects of noise and data imbalance, which limits our comprehensive comparison of method performance. The catastrophic inheritance problem was first fully defined in a study published on February 2, 2024 [1]. Currently, there is a lack of baselines for fine-tuning methods to address this problem, which is one of the reasons why we were unable to provide more comparisons in our experiments. Our work is the first to propose an efficient fine-tuning method that integrates customized regularization terms to alleviate the catastrophic inheritance problem caused by data imbalance and noise in pre-trained models. This innovative approach provides a new solution to the problem and paves the way for subsequent research.
>
> In the revised version, we have included methods such as DoRA, LoHA, and DyLoRA in the comparison, and analyzed their performance differences with BA-LoRA in detail. In addition, we have added comparisons with classic fine-tuning methods such as BitFit, HAdapter, PAdapter, and AdaLoRA to further demonstrate the advantages of BA-LoRA. We will continue to expand the experiments and include more baselines for comparison to fully verify the effectiveness and performance of BA-LoRA.

---

> ### Author Response · Authors · 2024-11-22
> **Response to Reviewer aJ7c (Part II)**
>
> > **Q3:** BA-LoRA requires specialized regularization terms for NLG and NLU tasks, which may raise generalization concerns if applied to tasks not covered in this paper, such as classification.
>
> **A3:** We understand your concerns about the generalization ability of BA-LoRA in classification tasks. First, it should be pointed out that LLM is mainly used to handle NLP tasks, especially natural language understanding (NLU) and natural language generation (NLG) tasks [18]. Considering the different requirements of these two types of tasks, we designed three regularization terms to optimize the fine-tuning process of pre-trained models and effectively alleviate the catastrophic inheritance problem.
>
> Specifically, in **NLU** and **NLG** tasks, we tailored regularization strategies for each task type:
>
> **In NLU tasks**:
> - **Consistency regularization**: Minimize the difference between the output of the pre-trained model and the fine-tuned model, maintain key information and enhance model stability.
> - **Diversity regularization**: Reduce the covariance of the fine-tuning output, increase diversity, and prevent overfitting.
> - **SVD regularization**: Focus on the main data components by maximizing the top k singular values ​​and improve the generalization ability of the model.
>
> **In NLG tasks**:
> - **Consistency regularization**: Use KL divergence to measure the difference in output distribution, ensure the consistency of generated text, and prevent information loss.
> - **Diversity regularization**: Maximize the entropy of the generated distribution, encourage diverse outputs, and thus improve text quality.
> - **SVD regularization**: By maximizing the top k singular values, focus on important information, reduce noise interference, and improve stability and diversity.
>
> To verify the effectiveness of these regularization terms, we conducted experiments on multiple tasks. BA-LoRA showed excellent performance in NLG tasks (such as mathematical reasoning tasks GSM8K, MATH, code generation tasks HumanEval, MBPP, and multi-turn dialogue tasks MT-Bench). At the same time, we also verified the effectiveness of BA-LoRA on **classification tasks (such as SST-2, MNLI, etc.)** of the GLUE benchmark of NLU. These experimental results fully show that the regularization strategy of BA-LoRA is not only applicable to NLU and NLG tasks, but also can effectively improve the performance of classification tasks.

---

> > ### Author Response · Authors · 2024-11-22
> > **Response to Reviewer aJ7c (Part III)**
> >
> > > **Q4:** As noted in the weakness section, this work highlights the catastrophic inheritance issue in the data, which other LoRAs do not fully address. However, the paper lacks a clear definition of the noise and imbalance issues it considers. While the authors claim BA-LoRA mitigates the adverse effects of catastrophic inheritance (Section 4.4.1, Line 379), based on Tables 1 and 2, the unclear definition makes it difficult to attribute the improvements solely to noise and imbalance mitigation. More examples are needed to demonstrate how BA-LoRA addresses specific noise and imbalance issues.
> >
> > **A4:** 1\. The definition of noise and imbalance problems is unclear
> >
> > In the paper, we made it clear that noise and imbalance problems mainly refer to **problems in pre-training data**. Specifically, pre-training data usually comes from large-scale data on the Internet, which is inevitably noisy and imbalanced. For example, advanced pre-training models such as GPT-4 and LLaMA also use pre-training data from a large amount of unlabeled Internet text, which is often not thoroughly cleaned and corrected. Therefore, in the subsequent fine-tuning stage, it is difficult for the model to effectively remove irrelevant or inaccurate information, which in turn affects the performance of the model in downstream tasks.
> >
> > We have clearly mentioned in the introduction of the paper (lines 41 to 50) that large-scale data brings problems such as redundancy, bias, and corruption. These problems will have a negative impact on the model during training, especially when the data is unbalanced or noisy, which may lead to a decline in model performance and weakened generalization ability. In addition, we have also supplemented the specific content of noise and imbalance problems in pre-training data in the appendix of the paper, including low-quality bias, redundancy, pollution, category imbalance, and harmful content bias.
> >
> > 2\. How ​​BA-LoRA mitigates the catastrophic inheritance problem and whether the performance improvement is entirely due to noise and imbalance mitigation
> >
> > We show in detail the effectiveness of BA-LoRA in mitigating noise and data imbalance in subsections **4.4.2** and **4.4.3** of the experimental section. In the **Noise Mitigation Analysis** experimental section, we selected BERT-L and GPT-2-XL models, which were pre-trained on BooksCorpus and Wikipedia (BERT-L) and WebText datasets (GPT-2-XL), respectively, where noise and imbalance are common. On the GLUE and GLUE-x benchmarks, BA-LoRA improves the performance of LoRA by 2.03% and 2.47% on BERT-L and GPT-2-XL models, respectively. These results demonstrate that BA-LoRA effectively mitigates the noise in the pre-training data through its regularization term and enhances the robustness and generalization ability of the model.
> >
> > In the **Imbalanced Data Mitigation Analysis** experiment section, we fine-tuned the BERT-L and GPT-2-XL models on the MNLI dataset and visualized the hidden features of the last layer using the t-SNE technique. The results show that BA-LoRA exhibits clearer class separation when dealing with data imbalance, especially the BERT-L model, which exhibits higher intra-class aggregation and clearer class boundaries. These analyses indicate that BA-LoRA can effectively mitigate the impact of unbalanced data in pre-training. Therefore, the performance improvements in Tables 1 and 2 can be attributed to the advantages of BA-LoRA in mitigating noise and imbalance problems.
> >
> > 3\. More examples are needed to show how BA-LoRA mitigates specific noise and imbalance problems
> >
> > We have demonstrated the specific effects of BA-LoRA in mitigating noise and data imbalance problems in the experiments in Sections **4.4.2** and **4.4.3**. The experimental results show that BA-LoRA significantly outperforms LoRA on the GLUE and GLUE-x benchmarks, verifying its effectiveness in mitigating the impact of noise on pre-training data. In the imbalanced data analysis, t-SNE visualization analysis further demonstrated the advantages of BA-LoRA in dealing with imbalanced data, especially in the BERT-L model, where BA-LoRA showed stronger ability to distinguish categories.
> >
> > In addition, in the Appendix section "Analysis of Different Scales and Model Types", we fine-tune 10 models using LoRA, PiSSA, and BA-LoRA methods, including LLaMA-2-7B, LLaMA-3-8B, Mistral-7B, Gemma-7B, Qwen1.5-7B, Yi-1.5-34B, etc. These models were then evaluated on the GSM8K and HumanEval tasks. The results show that BA-LoRA outperforms LoRA and PiSSA in all models and tasks, further demonstrating the effectiveness of BA-LoRA. In the Appendix section "Evaluating Performance at Different Levels", we compare the performance of LLaMA-2-7B and Mistral-7B-v0.1 models under different level settings, and evaluate their performance changes on the GSM8K and MATH datasets. The results confirm that BA-LoRA also shows its advantages under different rank settings.

---

> > > ### Author Response · Authors · 2024-11-22
> > > **Response to Reviewer aJ7c (Part Ⅳ)**
> > >
> > > > **Q5:** In Table1, it is not clear that what kind of metric is used to evaluate the performance in NLG tasks.
> > >
> > > **A5:** We have clarified the metrics used to evaluate NLG tasks in **Table 1** in the revised version, as follows:
> > > - **GSM8K** and **MATH** datasets (math problem solving tasks): We use **Accuracy** as the evaluation metric, which indicates the proportion of questions that the model correctly answers.
> > > - **HumanEval** and **MBPP** datasets (code generation tasks): We use **Pass@1**, which indicates the proportion of code that the model generates for the first time that passes all unit tests.
> > > - **MT-Bench** dataset (multi-turn dialogue generation task): We use **GPT-4 first-round scoring**, which is the quality evaluation of the initial response generated by the model by GPT-4.
> > >
> > > The updated table is as follows, clarifying the evaluation metrics for each dataset:
> > > | Datasets        | GSM8K     | MATH      | HumanEval | MBPP      | MT-Bench                        |
> > > |-----------------|-----------|-----------|-----------|-----------|---------------------------------|
> > > | Metric         | Accuracy  | Accuracy  | Pass@1    | Pass@1    | GPT-4 First Round Scoring|
> > >
> > >
> > > > **Q6:** In Section 4.3, the values of $\lambda_1$, $\lambda_2$, and $\lambda_3$ for NLG and NLU tasks are very small. Do these values contribute sufficiently to the overall loss function?
> > >
> > > **A6:** In the revised version of the paper, we additionally analyze why these regularization parameters have relatively small $\lambda$ values. First, smaller $\lambda$ values ​​help avoid over-constraining the model. Although these values ​​are small, they have a significant positive impact on the overall performance of the model in multiple experimental settings. To further illustrate this point, we added hyperparameter analysis experiments in the revised version, explicitly showing the impact of different $\lambda$ values ​​on the loss function and final model performance.
> > >
> > > The main goal of choosing appropriate regularization hyperparameters is to balance the generalization ability of the model and the regularization effect during training. We mainly followed the following steps when choosing regularization weights:
> > > 1. **Divide the validation set**: Split a part of the training data as the validation set to ensure that it does not participate in the training process and is specifically used to evaluate the performance of the model.
> > > 2. **Hyperparameter adjustment**: Within a reasonable range, we adjust the regularization weight by using a hyperparameter optimization algorithm (such as random search) and evaluate the performance of the model on the validation set under different settings. Reasonable ranges are usually set based on previous experience or preliminary experimental results.
> > > 3. **Select the best parameters**: Based on the performance indicators on the validation set, the regularization weight combination that can best improve the model's generalization ability is selected.
> > >
> > > > **Q7:** In sections 4.4.2 and 4.4.3, there are only results and analyses on NLU datasets. More results on both tasks are needed to better demonstrate the ability of BA-LoRA
> > >
> > > **A7:** First, in the Appendix "Analysis of Different Scales and Model Types", we fine-tuned 10 models using LoRA, PiSSA, and BA-LoRA methods, covering LLaMA-2-7B, LLaMA-2-13B, LLaMA-3-8B, LLaMA-3-70B, Mistral-7B-v0.1, Gemma-7B, Qwen1.5-7B, Yi-1.5-34B, and Mixed Expert Models (MoE): DeepSeek-MoE-16B and Mixtral-8x7B-v0.1. Subsequently, we evaluated the performance of these models on NLG tasks such as GSM8K and HumanEval. The results show that BA-LoRA outperforms LoRA and PiSSA on all models and tasks, highlighting its advantages in NLG tasks.
> > >
> > > In addition, in the Appendix "Evaluating the performance of different ranks", we compared the performance of LLaMA-2-7B and Mistral-7B-v0.1 models under different rank settings, focusing on evaluating their effects on the GSM8K and MATH NLG datasets. The results further confirm the effectiveness of BA-LoRA under various rank settings.
> > >
> > > In order to more comprehensively verify the effectiveness of our method, we will add more experimental results of the current state-of-the-art models on NLG tasks in the revised version.
> > >
> > > Once again, thank you for your valuable suggestions. Your feedback has been very helpful in refining the content of the paper. Should you have any further questions or suggestions, please do not hesitate to reach out.

---

> > > > ### Author Response · Authors · 2024-11-22
> > > > **Response to Reviewer aJ7c (Part Ⅴ)**
> > > >
> > > > [1] Chen H, Raj B, Xie X, et al. On catastrophic inheritance of large foundation models[J]. arXiv preprint arXiv:2402.01909, 2024.
> > > >
> > > > [2] Liu Z, He K. A Decade's Battle on Dataset Bias: Are We There Yet?[J]. arXiv preprint arXiv:2403.08632, 2024.
> > > >
> > > > [3] Yang Y, Singh A K, Elhoushi M, et al. Decoding data quality via synthetic corruptions: Embedding-guided pruning of code data[J]. arXiv preprint arXiv:2312.02418, 2023.
> > > >
> > > > [4] Chen H, Wang J, Shah A, et al. Understanding and mitigating the label noise in pre-training on downstream tasks[J]. arXiv preprint arXiv:2309.17002, 2023.
> > > >
> > > > [5] Elazar Y, Bhagia A, Magnusson I, et al. What's In My Big Data?[J]. arXiv preprint arXiv:2310.20707, 2023.
> > > >
> > > > [6] Carlini N, Ippolito D, Jagielski M, et al. Quantifying memorization across neural language models[J]. arXiv preprint arXiv:2202.07646, 2022.
> > > >
> > > > [7] Hernandez D, Brown T, Conerly T, et al. Scaling laws and interpretability of learning from repeated data[J]. arXiv preprint arXiv:2205.10487, 2022.
> > > >
> > > > [8] Fan L, Chen K, Krishnan D, et al. Scaling laws of synthetic images for model training... for now[C]//Proceedings of the IEEE/CVF Conference on Computer Vision and Pattern Recognition. 2024: 7382-7392.
> > > >
> > > > [9] Kreutzer J, Caswell I, Wang L, et al. Quality at a glance: An audit of web-crawled multilingual datasets[J]. Transactions of the Association for Computational Linguistics, 2022, 10: 50-72.
> > > >
> > > > [10] Roberts M, Thakur H, Herlihy C, et al. Data contamination through the lens of time[J]. arXiv preprint arXiv:2310.10628, 2023.
> > > >
> > > > [11] Schaeffer R. Pretraining on the test set is all you need[J]. arXiv preprint arXiv:2309.08632, 2023.
> > > >
> > > > [12] Jiang M, Liu K Z, Zhong M, et al. Investigating data contamination for pre-training language models[J]. arXiv preprint arXiv:2401.06059, 2024.
> > > >
> > > > [13] Xu H, Xie S, Tan X E, et al. Demystifying clip data[J]. arXiv preprint arXiv:2309.16671, 2023.
> > > >
> > > > [14] Zhu B, Tang K, Sun Q, et al. Generalized logit adjustment: Calibrating fine-tuned models by removing label bias in foundation models[J]. Advances in Neural Information Processing Systems, 2024, 36.
> > > >
> > > > [15] Parashar S, Lin Z, Liu T, et al. The Neglected Tails in Vision-Language Models[C]//Proceedings of the IEEE/CVF Conference on Computer Vision and Pattern Recognition. 2024: 12988-12997.
> > > >
> > > > [16] Zou A, Wang Z, Carlini N, et al. Universal and transferable adversarial attacks on aligned language models[J]. arXiv preprint arXiv:2307.15043, 2023.
> > > >
> > > > [17] Huang Y, Sun L, Wang H, et al. Trustllm: Trustworthiness in large language models[J]. arXiv preprint arXiv:2401.05561, 2024.
> > > >
> > > > [18] Karanikolas N, Manga E, Samaridi N, et al. Large language models versus natural language understanding and generation[C]//Proceedings of the 27th Pan-Hellenic Conference on Progress in Computing and Informatics. 2023: 278-290.

---

> > > > > ### Comment · Reviewer_aJ7c · 2024-11-23
> > > > > **Replies by Reviewer aj7c**
> > > > >
> > > > > 1. **Adding more baselines**
> > > > >
> > > > > By reading all replies to other reviewers, authors added several baseline comparisons with the proposed BA-LoRA. However, the improvement from BA-LoRA by comparing with other baselines such as AdaLoRA, DoRA, DyLoRA is incremental.
> > > > >
> > > > > 2. **Ablation studies**
> > > > >
> > > > > In addition, the ablation study shows that the performance differences on different tasks are very small by removing the proposed different regularization terms, which did not demonstrate the importance of those designed specific regularization terms for NLU and NLG tasks. Although authors claim that those designed regularization terms with very small hyperparameters have a significant positive impact on the overall performance of the model in multiple experimental settings, the new added baseline comparison experiments and ablation studies with incremental improvement and small differences actually cannot verify this statement.
> > > > >
> > > > > 3. **Revised manuscript**
> > > > >
> > > > > Although authors mentioned that they included those modifications in their revised manuscript, reviewer aj7c cannot find any updates in the submission. In addition, authors added so many new experiments in the replies, and it is questionable whether those new contents can fit into 10 pages of the main content.

---

> > > > > > ### Author Response · Authors · 2024-11-24
> > > > > > **Further Response to Reviewer aj7c**
> > > > > >
> > > > > > Thank you for your response. Regarding these three questions, our answers are as follows:
> > > > > >
> > > > > > > **Q1:** The improvement from BA-LoRA by comparing with other baselines such as AdaLoRA, DoRA, DyLoRA is incremental.
> > > > > >
> > > > > > **A1:** First, while the improvement of BA-LoRA over AdaLoRA, DoRA, and DyLoRA is incremental, its performance enhancement is consistent across a wide range of NLU and NLG tasks and datasets. This consistent improvement proves the effectiveness and practical value of the BA-LoRA approach.
> > > > > >
> > > > > > Second, BA-LoRA is a plug-and-play method that can enhance the effectiveness of LoRA and its variants. According to the results in Table 10 in Appendix C.3, the combination of LoRA and the three regularizers significantly improves the performance of the original LoRA by an average of 2.5 percentage points.
> > > > > >
> > > > > > Finally, our main goal is to alleviate the "catastrophic inheritance" phenomenon, e.g. the detrimental effects brought by noisy and imbalanced data from the pre-training dataset. The extensive experimental results in Sections 4.4.2 and 4.4.3 confirm the effectiveness of BA-LoRA in addressing this issue. In our opinion, the catastrophic inheritance for LLMs is a fundamental research topic in NLP,  we believe that address such fundamental problems can provide more insights into understanding LLM training and is way more important than solely pursuing performance gains.
> > > > > >
> > > > > >
> > > > > > > **Q2:** The ablation study shows that removing the proposed regularization terms results in very small performance differences across tasks, suggesting these terms may not significantly impact NLU and NLG tasks as claimed. Despite the authors' assertion that these regularization terms with small hyperparameters positively affect overall model performance, the minimal improvements observed in additional baseline comparisons and ablation studies do not substantiate this claim.
> > > > > >
> > > > > > **A2:** Thank you for your interest in our research and your valuable comments. In response to your query, we would like to clarify:
> > > > > >
> > > > > > First, in both NLU and NLG tasks, even smaller performance improvements can have a significant impact on practical applications. The results of the ablation study in Section 4.4.4 show that the joint use of the three regularization terms improves the model performance significantly. Although the gains of individual regularization terms are small, their concerted use can cumulatively enhance the performance and generalization of the model.
> > > > > >
> > > > > > Second, the main goal of this paper is to mitigate the negative impact of noisy and unbalanced pre-training data on downstream tasks, i.e., the “catastrophic inheritance” phenomenon. The experimental results in Sections 4.4.2 and 4.4.3 have confirmed the effectiveness of BA-LoRA in this regard. Improving the overall performance is not our primary goal; we focus on solving an important challenge in real-world applications, and the performance improvement is an added benefit in the process. We believe that solving critical problems in real applications is more important than simply pursuing performance improvements.
> > > > > >
> > > > > > Finally, general experiments may not be able to fully realize the effect of BA-LoRA. Therefore, we conduct two sets of extensive experiments in Sections 4.4.2 and 4.4.3 to demonstrate that BA-LoRA is effective in mitigating the detrimental effects of noisy and unbalanced data in the pre-training dataset. These improvements are important for the reliability and robustness of SFT process.
> > > > > >
> > > > > > > **Q3:** Despite the authors claiming to have included the requested modifications in their revised manuscript, reviewer aj7c cannot find any updates. Additionally, the numerous new experiments introduced in their responses raise concerns about fitting within the 10-page main content limit.
> > > > > >
> > > > > > **A3:** We are very sorry for the inconvenience caused by our late submission of the revised paper. We have submitted a revised manuscript with all additions and changes highlighted in blue. Regarding the 10-page content limit, we have contained the main content in the 10-page main text and placed the extensive experiments and analyses in the Appendix. According to the latest ICLR policy, there is no page limit for the Appendix.

---

> ### Comment · Reviewer_aJ7c · 2024-12-02
> **Replies by Reviewer aj7c**
>
> **This work’s experiments cannot prove their proposed method mitigates catastrophic inheritance.**
>
> The major issue of this work is that the proposed method and experiments cannot directly and conclusively prove the catastrophic inheritance is mitigated. As authors mentioned in the response to Reviewer M3Ew, **catastrophic inheritance contains many aspects, like noise, imbalanced data, repeated data, corrupted data, etc.[1]**, it cannot directly prove authors’ claim about catastrophic inheritance mitigation only via the regularization ablation study. In addition, those regularization terms are designed for consistency, diversity and generalizaibility, which are not explained by authors about why those regularization terms are closely related to catastrophic inheritance issue.
>
> In addition, **the improvement from BA-LoRA and ablation study regarding regularization terms is incremental, which has been agreed by authors during rebuttals**. Such small improvements cannot directly conclude that catastrophic inheritance is mitigated, becuase there are too many factors which will contribute such small improvements, such as high-quality finetuning data, different scheduler, different random seed, more finetuning steps, etc. This aligns with the **Reviewer M3Ew’s core question**, i.e., “**How can we conclusively prove that our method mitigates catastrophic inheritance?**”
>
> To summarize, the catastrophic inheritance mitigation argument proposed in this work seems to be overclaimed. The current experiments including observing performance gains and regularization ablation study are **not sufficient to prove the causation between applied regularization techniques and the mitigation of catastrophic inheritance**.
>
> Therefore, I will remain my score unchanged.

---

> > ### Author Response · Authors · 2024-12-02
> > **Further Response to Reviewer aj7c (Part II)**
> >
> > Thank you for your continued feedback and for sharing your concerns regarding our work. We appreciate the opportunity to clarify and address the points you've raised.
> >
> > > **Q1:** *Evidence of Mitigating Catastrophic Inheritance*
> >
> > **A1:** We understand your concern about conclusively demonstrating that BA-LoRA mitigates catastrophic inheritance. In our paper, we conducted two extensive experiments specifically designed to assess BA-LoRA's effectiveness in this regard:
> >
> > 1. **Noise Mitigation Analysis (Section 4.4.2):** We evaluated BA-LoRA's ability to handle noisy pre-training data by fine-tuning models on datasets known to contain noise and observing improvements over baseline methods. The results showed that BA-LoRA effectively reduces the adverse effects of noise, which is a key aspect of catastrophic inheritance.
> >
> > 2. **Imbalanced Data Mitigation Analysis (Section 4.4.3):** We analyzed BA-LoRA's effectiveness in dealing with imbalanced data distributions. By visualizing the hidden features using t-SNE, we demonstrated that BA-LoRA achieves clearer class separations than other methods, indicating better handling of data imbalance—another critical facet of catastrophic inheritance.
> >
> > These experiments provide empirical evidence that BA-LoRA can alleviate specific issues associated with catastrophic inheritance. We have detailed these analyses in the paper and have mentioned them in our previous responses.
> >
> > > **Q2:** *Relation Between Regularization Terms and Catastrophic Inheritance*
> >
> > **A2:** While the regularization terms—Consistency Regularization, Diversity Regularization, and Singular Value Decomposition Regularization—are named based on their immediate functions, their design is motivated by addressing specific aspects of catastrophic inheritance:
> >
> > - **Consistency Regularization** aims to reduce the impact of noise and corruption in pre-training data by ensuring the fine-tuned model's outputs remain consistent with the valuable information learned during pre-training.
> >
> > - **Diversity Regularization** encourages the model to explore a broader solution space, which helps mitigate issues arising from data imbalance and prevents overfitting to dominant patterns in the data.
> >
> > - **Singular Value Decomposition Regularization** focuses the model on the most significant components of the data, thereby reducing the influence of noise and redundancy inherent in the pre-training dataset.
> >
> > This motivation is explicitly discussed in the Method section (Section 3.2) of our paper, where we highlight how these regularizers contribute to alleviating different aspects of catastrophic inheritance.
> >
> > > **Q3:** *Significance of Improvements*
> >
> > **A3:** Regarding the concern about the improvements being incremental, we would like to clarify that while individual regularization terms may contribute modest improvements, their combined effect leads to significant enhancements in certain tasks. For example, according to the results in Table 10 in Appendix C.3, the combination of LoRA and the three regularizers improves the performance of the original LoRA by an average of **2.5 percentage points**. This improvement is substantial, especially in the context of large-scale language models, where even small percentage increases can lead to meaningful performance gains.
> >
> > Furthermore, BA-LoRA significantly outperforms the baseline method on the NLG task corresponding to LLaMA-2-7B, with an average improvement of **1.84 percentage points**. These consistent improvements across a variety of tasks and models highlight the robustness and practical value of our method.
> >
> > We have mentioned that for a fair comparison, we strictly followed the experimental settings provided in PiSSA. All baselines and our BA-LoRA were evaluated under the same conditions, ensuring that the observed improvements were due to our method rather than external factors. We do not believe it is reasonable to question the fairness of our experiments by suggesting factors that were not present, such as incorporating extra high-quality fine-tuning data, different schedulers, random seeds, or additional fine-tuning steps.
> >
> > **Conclusion**
> >
> > We believe that the combination of our targeted regularization strategies and the empirical evidence from our noise mitigation analysis (Section 4.4.2), imbalanced data mitigation analysis(Section 4.4.3), and ablation studies(Section 4.4.4) substantiate our claim that BA-LoRA mitigates catastrophic inheritance. We appreciate your feedback, which has prompted us to reinforce the explanations and justifications in our paper.
> >
> > Thank you again for your thoughtful review. We hope our response addresses your concerns, and we are open to any further questions or suggestions you may have.

---

### Official Review · Reviewer_BEfp · 2024-11-04

**Soundness:** 3
**Presentation:** 3
**Contribution:** 3
**Rating:** 8
**Confidence:** 3

**Summary:**

This paper investigates the challenges associated with fine-tuning large language models (LLMs), particularly focusing on catastrophic forgetting and bias introduction. It introduces BA-LoRA, a method aimed at improving the adaptation of LLMs while minimizing computational costs. The authors highlight the issue of catastrophic inheritance, where models lose previous knowledge when trained on new tasks. BA-LoRA employs a low-rank adaptation technique, allowing for efficient parameter updates during fine-tuning. Experimental results demonstrate that this approach effectively mitigates forgetting and reduces bias in adapted models, outperforming traditional fine-tuning methods. The findings suggest that BA-LoRA represents a significant step forward in making LLM adaptation more efficient and reliable for various natural language processing applications.

**Strengths:**

A regularization formula for LoRA training was proposed for NLU and NLG tasks respectively, and excellent improvement was achieved in model performance.

**Weaknesses:**

1.Ablation experiments need to supplement the experimental results of  Ltask_NLG and  Ltask_NLU to enrich Figure 2.


2.According to the results in Figure 2, Ldr_NLG improves LlaMA-2-7B more significantly than Gemma-7B. Is the " while LDR_NLG had a notably strong impact on Gemma-7B." in the text inconsistent with the picture information?


3.Why can we conclude that 'BA-LoRA can effectively alleviate the impact of imbalanced data in pre-training' based on Figure 1 and the observation that 'BA-LoRA in sub-figures (b) and (d) shows clearer category separation'? What is the relationship between these two observations?

**Questions:**

See Weaknesses

---

> ### Author Response · Authors · 2024-11-22
> **Response to Reviewer BEfp (Part I)**
>
> > **Q1:** Ablation experiments need to supplement the experimental results of Ltask_NLG and Ltask_NLU to enrich Figure 2.
>
> **A1:** We have included the ablation results of $\mathcal{L} _ {\text{task-NLU}}$ in the revised version. Specifically, we first test the performance of the fine-tuned model without regularization term ($w/o Reg$). Then test individual regularization terms: $\mathcal{L} _ {\text{CR-NLU}}$, $\mathcal{L} _ {\text{DR-NLU}}$, $\mathcal{L} _ {\text{SVDR-NLU}}$. Finally, BA-LoRA containing three regularization terms is tested. The following table shows the ablation experimental results of $\mathcal{L}_{\text{task-NLU}}$, using the DeBERTa-v3-base model, GLUE fine-tuned dataset:
>
> | Method                     | MNLI   | SST-2  | MRPC  | CoLA  | QNLI  | QQP   | RTE   | SST-B | Avg    |
> |----------------------------|--------|--------|-------|-------|-------|-------|-------|-------|--------|
> | $\mathit{w/o}\ \mathrm{Reg}$ | 90.47  | 95.81  | 91.48 | 72.27 | 94.41 | 92.21 | 87.14 | 91.93 | 89.47  |
> | $\mathcal{L} _ {\mathrm{CR-NLU}}$ | 90.84  | 96.27  | 91.65 | 72.68 | 94.64 | 92.47 | 87.59 | 92.11 | 89.78  |
> | $\mathcal{L} _ {\mathrm{DR-NLU}}$ | 90.77  | 96.09  | 91.81 | 72.44 | 94.44 | 92.41 | 87.37 | 91.89 | 89.65  |
> | $\mathcal{L} _ {\mathrm{SVDR-NLU}}$ | 90.63  | 95.96  | 91.37 | 72.37 | 94.58 | 92.39 | 87.35 | 92.08 | 89.59  |
> | BA-LoRA                     | 90.92  | 96.25  | 91.83 | 72.79 | 94.84 | 92.59 | 87.87 | 92.15 | 89.91  |
>
> The experimental results show that various regularizations have a positive effect on improving the performance of the BA-LoRA method, further confirming the effectiveness of the regularization terms designed in BA-LoRA for NLG tasks.
>
> > **Q2:** According to the results in Figure 2, Ldr_NLG improves LlaMA-2-7B more significantly than Gemma-7B. Is the " while LDR_NLG had a notably strong impact on Gemma-7B." in the text inconsistent with the picture information?
>
> **A2:** In response to this issue, we would like to clarify that in the original text, '$\mathcal{L} _ {\text{DR-NLG}}$ has a significantly stronger effect on Gemma-7B' is relative to $\mathcal{L} _ { \text{CR-NLG}}$, not to LLaMA-2-7B.
>
> In fact, what we mean is that on Gemma-7B, $\mathcal{L} _ {\text{DR-NLG}}$ exhibits more obvious performance than $\mathcal{L} _ {\text{CR-NLG}}$ Performance improvements. Specifically:
> - **GSM8K Task:**
>   - **Without regularization:** 77.58%
>   - **With $\mathcal{L} _ {\text{CR-NLG}}$:** 77.82% (Improvement: +0.24%)
>   - **With $\mathcal{L} _ {\text{DR-NLG}}$:** 77.87% (Improvement: +0.29%)
>
> - **MATH Task:**
>   - **Without regularization:** 31.47%
>   - **With $\mathcal{L} _ {\text{CR-NLG}}$:** 31.84% (Improvement: +0.37%)
>   - **With $\mathcal{L} _ {\text{DR-NLG}}$:** 31.92% (Improvement: +0.45%)
>
> Therefore, we have revised the statement in the manuscript to: “On Gemma-7B, $\mathcal{L} _ {\text{DR-NLG}}$ achieves a larger performance improvement compared to $\mathcal{L} _ {\text{CR-NLG}}$. For example, on the GSM8K task, $\mathcal{L} _ {\text{DR-NLG}}$ improves by +0.29%, while $\mathcal{L} _ {\text{CR-NLG}}$ improves by +0.24%; on the MATH task, $\mathcal{L} _ {\text{DR-NLG}}$ improves by +0.45%, while $\mathcal{L }_ {\text{CR-NLG}}$ improves by +0.37%.”

---

> > ### Comment · Reviewer_BEfp · 2024-11-25
> >
> > Thank you for the detailed response and the inclusion of the additional experimental results. The provided ablation studies and clarifications address the concerns effectively. The revised manuscript with clarified statements and quantitative improvements enhances the clarity and rigor of the work. I appreciate the effort in refining the presentation and ensuring consistency with the experimental data.
> >
> > Best regards.

---

> > > ### Author Response · Authors · 2024-11-25
> > > **Further Response to Reviewer BEfp**
> > >
> > > Thank you so much for your kind and encouraging feedback. We're delighted to know that the additional experimental results and clarifications have effectively addressed your concerns. Your insightful comments have been invaluable in helping us refine our work.
> > >
> > > If it wouldn't be too much trouble, we kindly ask if you might consider updating your rating of our manuscript based on these improvements.

---

> > > > ### Author Response · Authors · 2024-11-27
> > > > **Official Comment by Authors**
> > > >
> > > > We sincerely appreciate your positive feedback and for raising your rating of our paper. We're glad our revisions met your expectations, and your constructive comments were invaluable in helping us improve our work.

---

> > > > > ### Comment · Reviewer_BEfp · 2024-12-02
> > > > >
> > > > > Thank you for your kind words and appreciation of our feedback. We are pleased to see that the revisions you made have significantly improved the quality of your paper. Your dedication and professionalism are commendable.
> > > > >
> > > > > We look forward to seeing more of your high-quality research in the future.

---

> ### Author Response · Authors · 2024-11-22
> **Response to Reviewer BEfp (Part II)**
>
> > **Q3:** Why can we conclude that 'BA-LoRA can effectively alleviate the impact of imbalanced data in pre-training' based on Figure 1 and the observation that 'BA-LoRA in sub-figures (b) and (d) shows clearer category separation'? What is the relationship between these two observations?
>
> **A3:** In order to further clarify the regularization mechanism of BA-LoRA and its role in alleviating the impact of imbalanced data, we conducted a more in-depth analysis.
>
> First, The specific design of BA-LoRA for NLU tasks is as follows:
> 1. **Consistency Regularization**: Minimize the difference between the output of the pre-trained and fine-tuned models to ensure that the key information in the pre-training stage is effectively retained, thereby improving the stability of the model.
> 2. **Diversity Regularization**: Minimize the covariance of the model output during fine-tuning, promote the diversity of the output space, and reduce the model's over-reliance on certain specific categories, thereby alleviating the bias caused by data imbalance.
> 3. **SVD Regularization**: By maximizing the first $k$ singular values ​​of the fine-tuned output matrix, the model's focus on important information is enhanced, the interference of noise and redundant information is reduced, and the model's ability to distinguish in imbalanced data is improved.
>
> Next, the t-SNE visualization results in Figure 1 intuitively demonstrate the effect of BA-LoRA in alleviating the impact of imbalanced data:
> - **Sub-figures (a) and (c)** (standard LoRA fine-tuning): The distributions between categories overlap significantly and the boundaries are blurred, indicating that the imbalanced pre-training data has a negative impact on the model classification effect.
> - **Sub-figures (b) and (d)** (BA-LoRA fine-tuning): Compared with standard LoRA, BA-LoRA shows clearer category separation and clearer boundaries, indicating that BA-LoRA more effectively improves the ability to distinguish categories.
>
> t-SNE is a dimensionality reduction technique used to map high-dimensional data to a low-dimensional space to facilitate the observation of the distribution between categories. In this experiment, the t-SNE visualization results after BA-LoRA fine-tuning show that the separation between categories is significantly enhanced, verifying that BA-LoRA achieves a clearer category division in the feature space. These analyses demonstrate the effectiveness of BA-LoRA in dealing with imbalanced data.
>
> Once again, thank you for your valuable suggestions. Your feedback has been very helpful in refining the content of the paper. Should you have any further questions or suggestions, please do not hesitate to reach out.

---

### Official Review · Reviewer_xKHH · 2024-11-04

**Soundness:** 2
**Presentation:** 3
**Contribution:** 2
**Rating:** 6
**Confidence:** 4

**Summary:**

The paper introduces Bias-Alleviating Low-Rank Adaptation (BA-LoRA) with three regularization terms: (1) a consistency regularizer, (2) a diversity regularizer, and (3) a singular value decomposition regularizer. They conduct experiments on natural language understanding and natural language generation tasks, and the results demonstrate that BA-LoRA outperforms LoRA and PiSSA.

**Strengths:**

Three regularization terms are proposed for LoRA to mitigate the detrimental effects of Catastrophic Inheritance from pre-training.

The paper conducts experiments on natural language understanding and natural language generation tasks with serveral benchmarks and LLMs.

**Weaknesses:**

The experiments are not sufficient, which only compare limited baselines, but ignore many relevant methods, such as DoRA, LoHA, DyLoRA, DeltaLoRA, which are mentioned in related works. The paper does not provide enough evidence to support its claim that BA-LoRA outperforms LoRA and its state-of-the-art variants.

In the experiments, according to Table 1 and 2, why Full FT has worse performance than PEFT method?

The discussion about why these three regularization terms can mitigate the detrimental effects of Catastrophic Inheritance is not sufficient.

In my option, these regularization terms should be universally applicable. That is to say, the regularization terms should be added to LoRA’s variants, and improve their performance. More experiments should be conducted for this.

In section 3, why PISSA is described in such detail, is it highly relevant to BA-LoRA?

**Questions:**

See the weaknesses.

---

> ### Author Response · Authors · 2024-11-22
> **Response to Reviewer xKHH (Part I)**
>
> > **Q1:** The experiments are limited, ignoring methods like DoRA, LoHA, DyLoRA, and DeltaLoRA, and lack sufficient evidence to support the claim that BA-LoRA outperforms LoRA and its variants.
>
> **A1:** We have added comparisons with methods such as DoRA [1], LoHA [2], and DyLoRA [3] in the revised version, and deeply analyzed their performance differences with BA-LoRA. In addition, we have added comparisons with classic fine-tuning methods such as BitFit [4], Hadapter [5], Padapter [6], and AdaLoRA [7], which fully demonstrate the advantages of BA-LoRA.
> Regarding DeltaLoRA, we are currently unable to make a direct comparison because the code of this method has not been released.
>
> Specifically, we choose DeBERTa-v3-base as the experimental model and use the GLUE benchmark dataset. The comparison results are shown in the table below. BA-LoRA outperforms all compared methods in all tasks, further verifying the performance advantage of BA-LoRA.
>
> | Methods    | MNLI  | SST-2 | MRPC | CoLA  | QNLI  | QQP   | RTE   | SST-B | Avg    |
> |------------|-------|-------|------|-------|-------|-------|-------|-------|--------|
> | Full FT    | 89.90 | 95.61 | 89.50 | 69.23 | 94.09 | 92.44 | 83.85 | 91.71 | 88.29  |
> | BitFit     | 89.37 | 94.84 | 87.75 | 66.96 | 92.24 | 88.41 | 78.70 | 91.35 | 86.20  |
> | HAdapter   | 90.13 | 95.53 | 89.95 | 68.64 | 94.11 | 91.91 | 84.48 | 91.48 | 88.28  |
> | PAdapter   | 90.33 | 95.61 | 89.46 | 68.77 | 94.29 | 92.04 | 85.20 | 91.60 | 88.41  |
> | LoHA       | 90.74 | 94.92 | 90.43 | 70.63 | 93.95 | 92.05 | 86.41 | 91.72 | 88.86  |
> | DyLoRA     | 90.97 | 95.21 | 91.45 | 70.79 | 94.08 | 92.29 | 86.57 | 91.86 | 89.15  |
> | DoRA       | 90.29 | 95.79 | 90.93 | 70.85 | 94.10 | 92.17 | 86.04 | 91.79 | 89.00  |
> | AdaLoRA    | 90.76 | 96.10 | 90.69 | 71.45 | 94.55 | 92.23 | 87.59 | 91.84 | 89.40  |
> | **BA-LoRA**| **90.92** | **96.25** | **91.83** | **72.79** | **94.84** | **92.59** | **87.87** | **92.15** | **89.91** |
>
>
> > **Q2:** Why does Full FT perform worse than the PEFT method in Tables 1 and 2?
>
> **A2:** Regarding the issue you mentioned in Tables 1 and 2, where the performance of the Full FT method is lower than that of the PEFT method, we have conducted further analysis. Previous studies have shown that when a large number of parameters are updated during fine-tuning, particularly when the model is faced with a large and diverse training dataset, overfitting is prone to occur [8][9]. Additionally, large-scale parameter updates can lead to catastrophic forgetting [10][11]. In contrast, the PEFT method helps control the complexity of the model by fine-tuning only a subset of parameters, thereby reducing the risk of catastrophic forgetting and improving generalization ability [12][13].
>
> Furthermore, recent studies [14][15][16][17] have indicated that when the rank value and other key hyperparameters in low-rank fine-tuning (such as scaling factor $\alpha$, learning rate lr, etc.) are appropriately set, the PEFT method can sometimes outperform the Full FT method.

---

> ### Author Response · Authors · 2024-11-22
> **Response to Reviewer xKHH (Part II)**
>
> > **Q3:** The discussion on how these three regularization terms mitigate Catastrophic Inheritance is insufficient.
>
> **A3:** We have discussed the role of regularization in mitigating catastrophic inheritance in detail in the original paper and explained the relevant concepts and methods in multiple sections.
>
> First, in the introduction, we mentioned that the increase in data volume brings multiple challenges, including information imbalance, noise, and corruption, which may have a negative impact on model training (lines 41 to 50). We emphasized how noise and bias weaken the generalization ability of the model and pointed out that the potential bias in the pre-training data may still exist during the fine-tuning process, affecting the final performance and application effect of the model.
>
> In the **method section** (Section 3.2), we detailed how to mitigate the negative impact of catastrophic inheritance through the designed regularization terms, especially in NLU and NLG tasks.
>
> **For NLU tasks:**
> 1. **Consistency Regularization**: By minimizing the difference between the outputs of the pre-trained and fine-tuned models, this regularization helps preserve critical information from the pre-training phase, thus preventing knowledge loss or distortion and enhancing model stability.
> 2. **Diversity Regularization**: By minimizing the covariance of the fine-tuned outputs, this regularization increases the diversity of the model, alleviating overfitting caused by noise or bias, and helping the model learn a more uniform feature distribution.
> 3. **SVD Regularization**: By maximizing the first k singular values of the output matrix, this regularization reinforces the model’s focus on the main data components, improving the model's generalization ability and resistance to interference.
>
> **For NLG tasks:**
> 1. **Consistency Regularization**: By measuring the difference between the output distributions of the fine-tuned and pre-trained models using Kullback-Leibler divergence, this regularization ensures consistency and coherence in the generated text, preventing information loss or distortion associated with catastrophic inheritance.
> 2. **Diversity Regularization**: By maximizing the entropy of the generation distribution, this regularization encourages the generation of more diverse outputs, avoiding repetition and improving the quality and richness of the generated text.
> 3. **SVD Regularization**: By maximizing the contribution of the first k singular values, this regularization helps the model focus on the most important information in the data, reducing noise interference and improving the stability and diversity of the generated text.
>
> In the **Experiments section**, we verify the effectiveness of these regularization terms through extensive experiments. BA-LoRA performs well in the GLUE and GLUE-X benchmarks, especially under the conditions of noise and data imbalance, proving its stability and reliability. Through comparative experiments and ablation studies, we further verify the effectiveness of these regularization terms in mitigating catastrophic inheritance both independently and in synergy.
>
> In addition, in the **Appendix**, we discuss the noise and bias issues in pre-training data and their impact in the **Background** section. In the **More Experiments** section, we first evaluate the performance of more models of different sizes and types, and then evaluate the performance of models of different ranks. These experiments expand the scope of our experiments and demonstrate the effectiveness of BA-LoRA. Finally, t-SNE visualization shows the effectiveness of BA-LoRA in handling imbalanced data.

---

> ### Author Response · Authors · 2024-11-22
> **Response to Reviewer xKHH (Part III)**
>
> > **Q4:** In my opinion, these regularization terms should be universally applicable, enhancing LoRA's variants, and more experiments are needed to confirm this.
>
> **A4:** We have added further experimental evaluations in the revised version. The following table shows the performance comparison of LoRA, DoRA, PiSSA, and BA-LoRA with and without regularization terms, where "Reg" refers to the three regularization terms designed for each NLG task:
>
> | Methods      | #Params | GSM8K     | MATH      | HumanEval | MBPP      | MT-Bench | Avg       |
> |--------------|---------|-----------|-----------|-----------|-----------|----------|-----------|
> | Full FT      | 6738M   | 49.05     | 7.22      | 21.34     | 35.59     | 4.91     | 23.62     |
> | LoRA         | 320M    | 42.47     | 5.60      | 17.03     | 31.48     | 4.62     | 20.24     |
> | LoRA + Reg   | 320M    | 49.26     | 6.84      | 20.16     | 32.75     | 4.75     | 22.75     |
> | DoRA         | 321M    | 42.12     | 6.28      | 16.97     | 22.07     | 4.53     | 18.46     |
> | PiSSA        | 320M    | 52.01     | 7.76      | 21.55     | 33.09     | 4.87     | 23.86     |
> | DoRA + Reg   | 321M    | 52.80     | 8.05      | 21.94     | 34.61     | 5.02     | 24.48     |
> | BA-LoRA      | 320M    | **53.83** | **9.13**  | **23.58** | **36.86** | **5.11** | **25.70** |
>
> As shown in the table above, incorporating regularization terms into either LoRA or DoRA significantly enhances the performance of LoRA variants across all tasks. This demonstrates that regularization is highly applicable to different LoRA variants. Furthermore, BA-LoRA, which combines PiSSA with regularization, achieves the best performance across various tasks and markedly improves the model's generalization ability.
>
> > **Q5:** In section 3, why is PISSA described in such detail, is it highly relevant to BA-LoRA?
>
> **A5:** Although PiSSA and BA-LoRA each have different focuses, PiSSA is a core component of our approach. When PiSSA is used in combination with regularization terms, the performance and training efficiency of the model can be significantly improved. Given the strong connection between PiSSA and BA-LoRA, we believe it is necessary to elucidate how PiSSA works and its interaction with BA-LoRA. In the revised version, we have simplified some details to avoid redundancy while ensuring that key content is highlighted.
>
> Once again, thank you for your valuable suggestions. Your feedback has been very helpful in refining the content of the paper. Should you have any further questions or suggestions, please do not hesitate to reach out.

---

> ### Author Response · Authors · 2024-11-22
> **Response to Reviewer xKHH (Part Ⅳ)**
>
> **References**
>
> [1] Liu S Y, Wang C Y, Yin H, et al. Dora: Weight-decomposed low-rank adaptation[J]. arXiv preprint arXiv:2402.09353, 2024.
>
> [2] Hyeon-Woo N, Ye-Bin M, Oh T H. Fedpara: Low-rank hadamard product for communication-efficient federated learning[J]. arXiv preprint arXiv:2108.06098, 2021.
>
> [3] Valipour M, Rezagholizadeh M, Kobyzev I, et al. Dylora: Parameter efficient tuning of pre-trained models using dynamic search-free low-rank adaptation[J]. arXiv preprint arXiv:2210.07558, 2022.
>
> [4] Zaken E B, Ravfogel S, Goldberg Y. Bitfit: Simple parameter-efficient fine-tuning for transformer-based masked language-models[J]. arXiv preprint arXiv:2106.10199, 2021.
>
> [5] Houlsby N, Giurgiu A, Jastrzebski S, et al. Parameter-efficient transfer learning for NLP[C]//International conference on machine learning. PMLR, 2019: 2790-2799.
>
> [6] Pfeiffer J, Kamath A, Rücklé A, et al. Adapterfusion: Non-destructive task composition for transfer learning[J]. arXiv preprint arXiv:2005.00247, 2020.
>
> [7] Zhang Q, Chen M, Bukharin A, et al. AdaLoRA: Adaptive budget allocation for parameter-efficient fine-tuning[J]. arXiv preprint arXiv:2303.10512, 2023.
>
> [8] Shi J X, Wei T, Zhou Z, et al. Long-Tail Learning with Foundation Model: Heavy Fine-Tuning Hurts[C]//Forty-first International Conference on Machine Learning. 2024.
>
> [9] Jiang H, He P, Chen W, et al. Smart: Robust and efficient fine-tuning for pre-trained natural language models through principled regularized optimization[J]. arXiv preprint arXiv:1911.03437, 2019.
>
> [10] Zhai Y, Tong S, Li X, et al. Investigating the catastrophic forgetting in multimodal large language model fine-tuning[C]//Conference on Parsimony and Learning. PMLR, 2024: 202-227.
>
> [11] Zhai Y, Tong S, Li X, et al. Investigating the catastrophic forgetting in multimodal large language model fine-tuning[C]//Conference on Parsimony and Learning. PMLR, 2024: 202-227.
>
> [12] Mao Y, Ge Y, Fan Y, et al. A survey on lora of large language models[J]. arXiv preprint arXiv:2407.11046, 2024.
>
> [13] Fu Z, Yang H, So A M C, et al. On the effectiveness of parameter-efficient fine-tuning[C]//Proceedings of the AAAI conference on artificial intelligence. 2023, 37(11): 12799-12807.
>
> [14] Bałazy K, Banaei M, Aberer K, et al. LoRA-XS: Low-Rank Adaptation with Extremely Small Number of Parameters[J]. arXiv preprint arXiv:2405.17604, 2024.
>
> [15] Meng F, Wang Z, Zhang M. Pissa: Principal singular values and singular vectors adaptation of large language models[J]. arXiv preprint arXiv:2404.02948, 2024.
>
> [16] Wang S, Yu L, Li J. LoRA-GA: Low-Rank Adaptation with Gradient Approximation[J]. arXiv preprint arXiv:2407.05000, 2024.
>
> [17] Yang Y, Li X, Zhou Z, et al. CorDA: Context-Oriented Decomposition Adaptation of Large Language Models for Task-Aware Parameter-Efficient Fine-tuning[C]//The Thirty-eighth Annual Conference on Neural Information Processing Systems.

---

> ### Comment · Area_Chair_2poc · 2024-12-02
>
> Dear Reviewer,
>
> I noticed that you haven't yet responded to the author's rebuttal. As the deadline of discussion period is approaching, could you please review their responses and provide your feedback? Thank you for your prompt attention to this matter.
>
> Area Chair

---

> > ### Comment · Reviewer_xKHH · 2024-12-02
> >
> > Thank the authors for the detailed responses and the additional experiments. I also read reviews and replies from other reviewers. I will update the rating for the paper.  Unresolved concerns: in order to provide sufficient evidence to support the claim that BA-LoRA outperforms LoRA and its variants, the experiments on NLG tasks should also be included.

---

> > > ### Author Response · Authors · 2024-12-02
> > > **Further Response to Reviewer xKHH(Part I)**
> > >
> > > We sincerely appreciate your positive feedback and are glad our revisions have addressed your concerns. Your constructive comments have been invaluable in improving our work.
> > >
> > > Regarding your suggestion about including experiments on NLG tasks, we are currently conducting these experiments and will include the results in the next version of the paper.
> > >
> > > Thank you again for your valuable time and consideration. We hope that the forthcoming results will further demonstrate the effectiveness of BA-LoRA.

---

### Author Response · Authors · 2024-11-29
**Kind Reminder**

Dear Reviewers,

We would like to extend our heartfelt gratitude for your thoughtful and constructive feedback during the review process. Your insights have been invaluable in enhancing the quality of our manuscript. In response to your comments, we have made the necessary revisions, and for your convenience, all changes have been highlighted in blue.

We kindly hope that you will find the revisions satisfactory and would greatly appreciate it if you could review the updated manuscript at your earliest convenience. If you have any additional comments or require further clarification, please feel free to let us know. Your continued input is highly valued.

Once again, thank you for your time, expertise, and consideration.

Best regards,

---

### Meta-Review · Area_Chair_2poc · 2024-12-19

**Metareview:**

This paper proposes BA-LoRA to mitigate catastrophic inheritance in LLMs through three regularization terms (consistency, diversity, SVD regularization). The authors claim BA-LoRA could outperform (vanilla) LoRA and other parameter-efficient fine-tuning methods, and effectively reduce the bias propagation from pretraining data.

Strengths: This paper addresses an important challenge - bias mitigation during LLM fine-tuning. Both quantitative and qualitative (t-SNE visualization) analysis have been conducted.

Weaknesses: The main issue raised by the reviewers is that the experimental results cannot support the main claim. While performance improvements are shown, the causation between these improvements and the reduction in catastrophic inheritance is not clearly established (and defined). The experimental setup also raises some concerns, as baseline methods lack standard regularization methods. Therefore, I recommend to reject this paper.

**Additional Comments On Reviewer Discussion:**

The reviewers and authors had multi-round interactions during the discussion period. While the authors were responsive and provided detailed explanations, they couldn't fully address the concern about proving their method specifically tackles catastrophic inheritance. This concern has been intensively discussed again during the internal discussion period. Therefore, I would suggest that the authors reconsider their major claim with the experimental setup and design.

---

### Decision · Program_Chairs · 2025-01-22

Reject